# Crimean–Congo haemorrhagic fever virus uses LDLR to bind and enter host cells

Vanessa M. Monteil [1,2], Shane C. Wright [3,23], Matheus Dyczynski[4,5,23],
Max J. Kellner[6,7], Sofia Appelberg [2], Sebastian W. Platzer [6,7], Ahmed Ibrahim[8],
Hyesoo Kwon [1,9], Ioannis Pittarokoilis [3], Mattia Mirandola[10],
Georg Michlits [5], Stephanie Devignot [1,2], Elizabeth Elder [9],
Samir Abdurahman[2], Sándor Bereczky[2], Binnur Bagci[11], Sonia Youhanna[3],
Teodor Aastrup[8], Volker M. Lauschke [3,12,13], Cristiano Salata [10],
Nazif Elaldi [14], Friedemann Weber [15], Nuria Monserrat [16,17,18],
David W. Hawman [19], Heinz Feldmann [19], Moritz Horn[4,5],
Josef M. Penninger [6,20,21,22] ✉ & Ali Mirazimi [1,2,9] ✉

Climate change and population densities accelerated transmission of highly pathogenic viruses to humans, including the Crimean–Congo haemorrhagic fever virus (CCHFV). Here we report that the Low Density Lipoprotein Receptor (LDLR) is a critical receptor for CCHFV cell entry, playing a vital role in CCHFV infection in cell culture and blood vessel organoids. The interaction between CCHFV and LDLR is highly specific, with other members of the LDLR protein family failing to bind to or neutralize the virus. Biosensor experiments demonstrate that LDLR specifically binds the surface glycoproteins of CCHFV. Importantly, mice lacking LDLR exhibit a delay in CCHFV-induced disease. Furthermore, we identified the presence of Apolipoprotein E (ApoE) on CCHFV particles. Our findings highlight the essential role of LDLR in CCHFV infection, irrespective of ApoE presence, when the virus is produced in tick cells. This discovery holds profound implications for the development of future therapies against CCHFV.

Crimean–Congo haemorrhagic fever virus (CCHFV), the causative agent of Crimean–Congo haemorrhagic fever (CCHF), is an emerging infectious agent that can lead to severe disease and has a mortality of up to 40% (World Health organization). Currently, there are no preventive or effective therapeutic measures available against CCHFV, which is listed as a key priority in the WHO's R&D Blueprint list of infectious agents with epidemic or pandemic potential. CCHF is a widespread haemorrhagic fever, which is endemic in certain regions of Africa and Asia, and is also spreading in Europe[1]. CCHFV is a tick-borne pathogen, transmitted by ticks of the *Hyalomma* genus, which can also be transmitted between humans via interpersonal contact. Because of global warming, the geographic zones where this tick vector can reside are expanding[2–4], thereby multiplying the risk of spreading by human transmission. The lack of approved interventions against CCHFV, either prophylactic or

therapeutic, combined with its increasing topographical range, constitutes a serious public health threat for many world regions.

Despite intensive research, much of the molecular pathogenesis of CCHFV is still unknown, including the identity of its receptor(s). Previous studies have shown that CCHFV enters cells through clathrin-mediated endocytosis[5,6] and uses the endosomal pathway to release viral RNA strands[7]. In vitro, many different cell types can be infected with CCHFV[8–12], suggesting the existence of either a widely distributed receptor or several redundant entry receptors. Of note, while nucleolin[13] and DC-SIGN[14] have been suggested as important entry factors, these data have not been confirmed and cannot explain cell entry or the broad cell tropism.

Here we report the identification of the Low Density Lipoprotein Receptor (LDLR) as an important in vitro and in vivo receptor for

CCHFV, including patient isolates, patient serum containing virus as well as virus produced on tick cells. We also demonstrate that LDLR specifically binds to Gn-Gc of CCHFV. In addition, we demonstrate that the knockout of *Ldlr* in mice is able to delay the disease. Finally, we highlight the importance of the cellular proteins located at the surface of the virus in virus entry.

## Results

### Haploid cell screening pinpoints *Ldlr*

Genome-wide screening methods have facilitated and accelerated the identification and characterization of host genes involved in infectious diseases. In particular, CRISPR/Cas9-based screens[15,16] and insertional mutagenesis in haploid cell systems[17,18] have enabled the discovery of receptors and intracellular host factors for various virus infections, including Ebola[19–21], Lassa[22,23] and SARS-CoV-2 (refs. [16,21]). With only a single copy of the genome, haploid cells offer direct translation of introduced genetic changes to a respective phenotype[24,25]. Combining haploid cells with genome saturating chemical mutagenesis using *N*-ethyl-*N*-nitrosourea (ENU)[26], we have developed an unbiased screening system that interrogates single nucleotide variants for their relevance in viral infections.

To identify host factors involved in CCHFV infections, we performed resistant screens using ENU-mutagenized murine haploid cells (AN3-12) with a viral RNA replication competent vesicular stomatitis virus, pseudotyped with the glycoproteins of the Crimean–Congo haemorrhagic fever virus (VSV-CCHF_G) (Extended Data Fig. 1a). This virus lacks the region coding for any glycoproteins and therefore produces non-infectious particles unless reconstituted with a novel surface glycoprotein, that is, in our screen, with the glycoproteins coded by the M segment of CCHFV. To validate the functionality of the glycoprotein complex in our pseudovirus under our experimental conditions, we conducted a seroneutralization test. Our observations revealed a dose-dependent inhibition of infection, confirming that the pseudovirus entry was mediated by the glycoproteins (Extended Data Fig. 1b). Infection with VSV-CCHF_G efficiently killed the haploid cells. Genome-wide, single amino acid mutagenesis in haploid cells resulted in the emergence of resistant colonies to VSV-CCHF_G-mediated killing (Extended Data Fig. 1c). These resistant colonies were individually selected, expanded and rescreened using the infectious CCHFV IbAr10200 laboratory strain (Extended Data Fig. 1d). Subsequently, whole-exome sequencing was conducted on the resistant clones. Three clones that showed nearly 100% resistance to CCHFV, namely, clones 5, 8 and 10 (Extended Data Fig. 1d), displayed mutations in the gene encoding Low Density Lipoprotein Receptor (*Ldlr*) (Extended Data Table 1). The mutations occurred at different locations in the *Ldlr* gene, probably resulting in gene knockout. Of note, we did not observe protein coding mutations in the other resistant colonies, suggesting that these mutations might be in regulatory gene regions. Because of this, we focused on *Ldlr*. These data identify *Ldlr* as a candidate gene for CCHFV infections.

### Validation of LDLR in CCHFV infections

To verify the role of LDLR in CCHFV entry, we first assessed *Ldlr* mutant haploid mouse embryonic cells and their respective Haplobank wild-type sister clones, as described previously[27]. These *Ldlr*-knockout cells and wild-type sister cells were infected with VSV as a positive control, as VSV is known to use LDLR as a receptor[28], and with VSV-CCHF_G and CCHFV. These murine *Ldlr*-knockout cells displayed more than a 90% decrease in infection rates compared with the wild-type cells (Fig. 1a). To investigate whether LDLR can also act as receptor for other bunyaviruses, we tested Rift Valley fever virus (RVFV), which has previously been shown to interact with the LDLR family member LDL Receptor Protein 1 (LRP1)[29,30]. In contrast to VSV-CCHF_G and CCHFV challenge, RVFV infection was not affected by the knockout of *Ldlr* as assessed by quantitative PCR with reverse transcription (RT–qPCR) (Fig. 1a).

Since these haploid cells are murine cells, it was paramount to confirm our results using African green monkey kidney epithelial cells (Vero) and human lung epithelial A549 cells, both being susceptible to CCHFV infection. We mutated the *LDLR* gene in these cell lines using CRISPR/Cas9 editing (Extended Data Fig. 2). Both *LDLR* mutant A549 as well as Vero cells showed a marked reduction in CCHFV infection compared with their respective LDLR-expressing control cells, determined by viral RNA detection at 24 h post infection (Fig. 1b,c). As a control, RVFV infections were unaffected in *LDLR* mutant A549 as well as in *LDLR* mutant Vero cells (Fig. 1d,e). Immunofluorescence data of wild-type and *LDLR* KO CCHFV-infected cells are presented in Extended Data Fig. 4a. These genetic deletion data validate the role of LDLR in CCHFV infections across species.

### Gc binds to LDLR and induces endocytosis

To test whether the LDLR can directly bind glycoproteins of CCHFV, we developed a bioluminescence resonance energy transfer (BRET) assay to assess receptor–ligand interactions[31]. To achieve this, we genetically engineered and expressed an LDLR that carries an N-terminal bioluminescent probe (NanoLuc) in HEK293 cells. Addition of fluorescent ligands then allows to measure direct interaction through BRET. BODIPY-FL-labelled LDL (LDL being the natural ligand of LDLR) resulted in the expected concentration-dependent increase in BRET signal in cells expressing Nluc-LDLR (Fig. 2a and Extended Data Fig. 3). As expected, an excess of unlabelled LDL outcompeted the labelled LDL for receptor binding as indicated by the lower BRET response. Likewise, the addition of the CCHFV glycoprotein Gc (10 µg ml$^{-1}$), but not Gn (10 µg ml$^{-1}$) decreased BRET, suggesting that Gc, but not Gn, directly bind to LDLR. Confirming this interaction, the addition of soluble BODIPY-FL-labelled Gc also resulted in a dose-dependent increase in BRET at LDLR (Extended Data Fig. 3), suggesting that Gc directly binds to LDLR in living cells. Given that Gc and Gn form heterodimers, we postulated that Gn may affect the binding behaviour of Gc. Interestingly, at concentrations where no binding was observed with any of the glycoproteins individually (1 µg ml$^{-1}$), adding Gn and Gc together resulted in a synergistic effect on the binding to LDLR highlighted by the decrease in BRET observed for the combination of Gc and Gn at 1 µg ml$^{-1}$ (Fig. 2a). Using a quartz crystal microbalance (QCM) with immobilized extracellular domains of LDLR, we measured affinity and binding kinetics. In line with the BRET data, LDL and Gc, but not Gn, resulted in dose-dependent increases in frequency, indicative of binding, with affinities of 3 and 3.3 nM, respectively. Notably, for the combination of Gc and Gn, a longer kinetic was required to establish the off rate and revealed an affinity that was an order of magnitude stronger than that of Gc alone (272 pM) (Fig. 2b,c). In contrast, the binding affinity of GP38, a secreted CCHFV glycoprotein (GP38) of unknown function that is the target of protective antibodies, was 1,000-fold lower (0.18 µM) than the affinities of LDL or Gc for LDLR. No binding was observed upon addition of the glycoprotein from the related Toscana virus (Extended Data Fig. 3b). All affinity constants are presented in Extended Data Table 2.

Clearance of circulating LDL from the bloodstream and subsequent LDL hydrolysis is achieved through its uptake by the LDLR via receptor-mediated endocytosis[32]. To investigate the mechanism by which LDLR facilitates CCHFV infection, we next engineered LDLR to express RlucII, a bioluminescent donor, at its C terminus. A marker for early endosomes, the FYVE domain of the human endofin[33], tagged with a fluorescent acceptor rGFP, was then co-expressed with LDLR-RlucII to measure the degree of internalized LDLR (Fig. 2d). Exposure of cells to LDL led to a concentration-dependent increase in BRET between LDLR-RlucII and rGFP-FYVE, confirming that LDLR traffics to endosomes upon binding to LDL (Fig. 2d). Importantly, addition of the CCHFV surface glycoprotein Gc, but not Gn, triggered endocytosis and enrichment of LDLR in early endosomes (Fig. 2d). In contrast, the recombinant receptor binding domain (RBD) from SARS-CoV-2 was

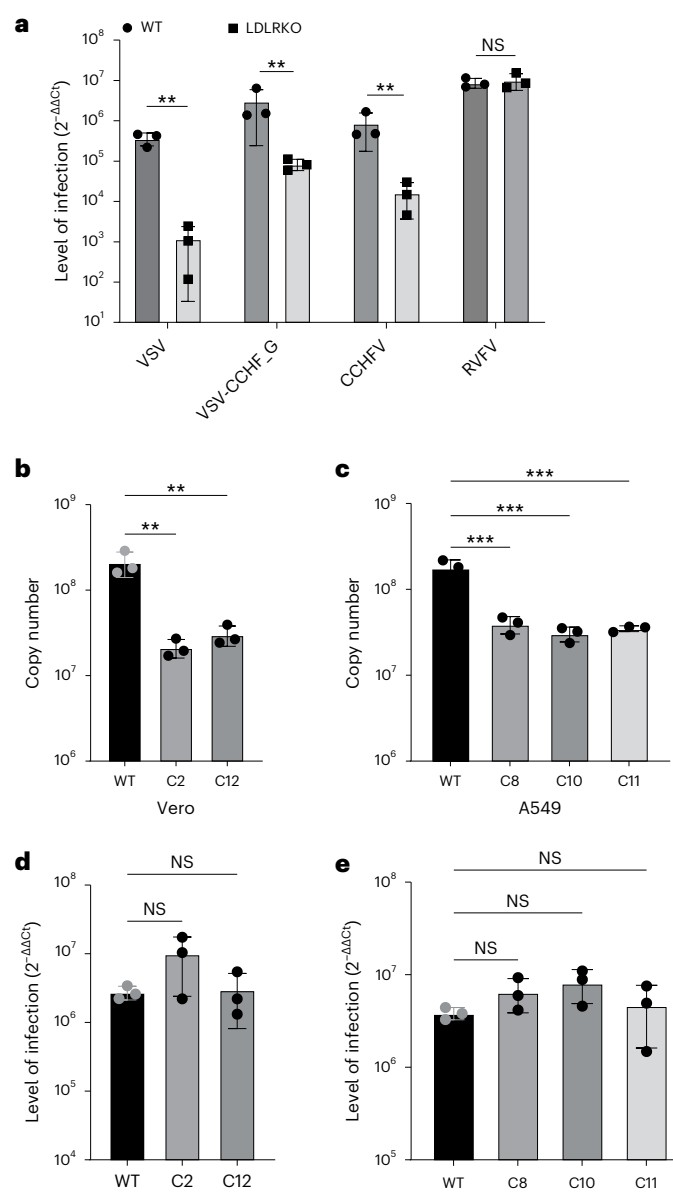

**Fig. 1 | CCHFV infections in *Ldlr*-knockout cells. a**, Levels of infection in control wild-type AN3-12 haploid and sister knockout (KO) cells infected with VSV, VSV-CCHF_G, CCHFV IbAr10200 and RVFV (MOI 0.1, 48 h post infection (h.p.i.)). Level of infection was assessed by RT–qPCR for viral and RNase P RNA. **b**, Levels of infection of IbAr10200 CCHFV in wild-type (WT) and two different *LDLR* KO (clones C2 and C12) Vero cells and **c**, in three different clones of *LDLR* KO (clones C8, C10 and C11) A549 cells. **d**, Levels of infection of RVFV in wild-type and two different *LDLR* KO (clones C2 and C12) Vero cells and **e**, in three different clones of *LDLR* KO (clones C8, C10 and C11) A549 cells. All mutant clones in **b**–**e** were generated using CRISPR/cas9 (Extended Data Fig. 3). Mutant haploid clones were from our previously reported Haplobank. All infections of diploid cells were done at an MOI of 0.1 for 24 h. Data are mean ± s.d. of *n* = 3 independent experiments. *P* values were calculated using two-sided unpaired Student's *t*-test (Fig. 2a) and one-way ANOVA (Fig. 2b–e). **$P < 0.01$, ***$P < 0.001$, NS $P > 0.05$. Exact *P* values are available in.

unable to elicit LDLR endocytosis (Extended Data Fig. 3c), demonstrating the specificity of LDLR internalization upon CCHFV Gc binding. Taken together, these results demonstrate that CCHFV Gc directly binds to human LDLR and exploits its endocytic pathway to infect cells.

Thus, we demonstrated that the recombinant Gc, and especially Gn-Gc, can bind to LDLR. Furthermore, to validate this interaction for

the virus, we conducted a competition assay using CCHFV and LDL, the natural ligand for LDLR that binds to LDLR present on the cell surface. As depicted in Fig. 2e, LDL effectively competed with CCHFV infection in a dose-dependent manner. bovine serum albumin (BSA) was used as a control and showed no impact on CCHFV infection (Fig. 2f).

## Soluble LDLR blocks CCHFV infections

Soluble receptor decoys have been successfully developed to block infections; for instance, soluble ACE2 has been used to effectively block SARS-CoV-2 infections[34–36]. To investigate the potential of soluble LDLR (sLDLR) as a molecular decoy to inhibit CCHFV infections, we added varying concentrations of sLDLR to VSV-CCHF_G or CCHFV for 30 min before cell infection. sLDLR is a one-chain LDLR Ala22Arg788 fragment. VSV was used as a positive control due to its known reliance on LDLR, but also VLDLR (Very Low Density Lipoprotein Receptor), as its receptors[28], and RVFV which binds to LRP1 was used as a negative control. Because the VSV life cycle in cells, from entry to new virus egress, is extremely rapid[37], VSV and VSV-CCHF_G infected cells have to be assessed at early timepoints post infection to avoid a second round of infection. Thus, at 6 h post infection for VSV and VSV-CCHF_G, and 24 h post infection for CCHFV and RVFV, the cells were collected and the levels of infection determined by RT–qPCR. Interestingly, sLDLR blocked the infection of rVSV-CCHF (Fig. 3a) and, importantly, CCHFV (Fig. 3b) in a dose-dependent manner. As expected, sLDLR was also able to inhibit VSV (Fig. 3c) but not RVFV infections (Fig. 3d).

Furthermore, we explored the potential of another member of the LDLR family, VLDLR, to hinder CCHFV and control VSV infections using soluble decoys. The soluble VLDLR (sVLDLR) utilized in our experiments is a single-chain fragment spanning from Thr25 to Ser797. In contrast to sLDLR, sVLDLR decoys demonstrated no effect against CCHFV infections (Fig. 3e) while remaining active in countering VSV infection (Fig. 3f). Immunofluorescence data depicting LDLR-treated and untreated CCHFV-infected cells are presented in Extended Data Fig. 4b. These data indicate that soluble LDLR can partially prevent CCHFV infections.

## Infection of blood vessel organoids

Our data so far showed that LDLR can function as a receptor for CCHFV infection. To further investigate the role of LDLRs in CCHFV infections in a human relevant model, we generated human blood vessel organoids using the induced pluripotent stem-cells (iPSC) line NC8 (ref. 38). Of note, blood vessels are key target cells for viral tropism involved in haemorrhaging, and endothelial cells are a target for CCHFV[39,40]. We then deleted *LDLR* in the iPSCs using CRISPR/Cas9-mediated genome editing. Knockout iPSCs for *LDLR* were validated by flow cytometry (Extended Data Fig. 5a–e). For infection, we generated *LDLR* mutant and wild-type blood vessel organoids (BVOs) containing self-organizing bona fide capillaries formed by pericytes and endothelial cells[38]. During these experiments, we realized that the culture conditions of these BVOs are sometimes detrimental to infection as the organoids are grown in a collagen/matrigel matrix that makes the cells less accessible to the virus. We therefore disaggregated the BVOs containing mature human capillaries and continued culture of the pericytes and endothelial cells as monolayers in collagen-coated flasks (Fig. 4a). These cultures were subsequently infected with CCHFV. Knockout of *LDLR* using two different mutant iPSC clones resulted in a significant reduction in CCHFV infections, detected at 1 and 3 days post infection (Fig. 4b). These data show that LDLR is also an important factor for CCHFV infections of human blood vessels.

## LDLR KO delays the disease in mice

We next investigated whether the absence of LDLR can protect against in vivo CCHFV infections and CCHF disease manifestations using C57BL/6J wild-type and *Ldlr*[−/−] mice. C57BL/6J mice are naturally resistant to CCHF[41], but blockade or knockout of IFNα receptors render these

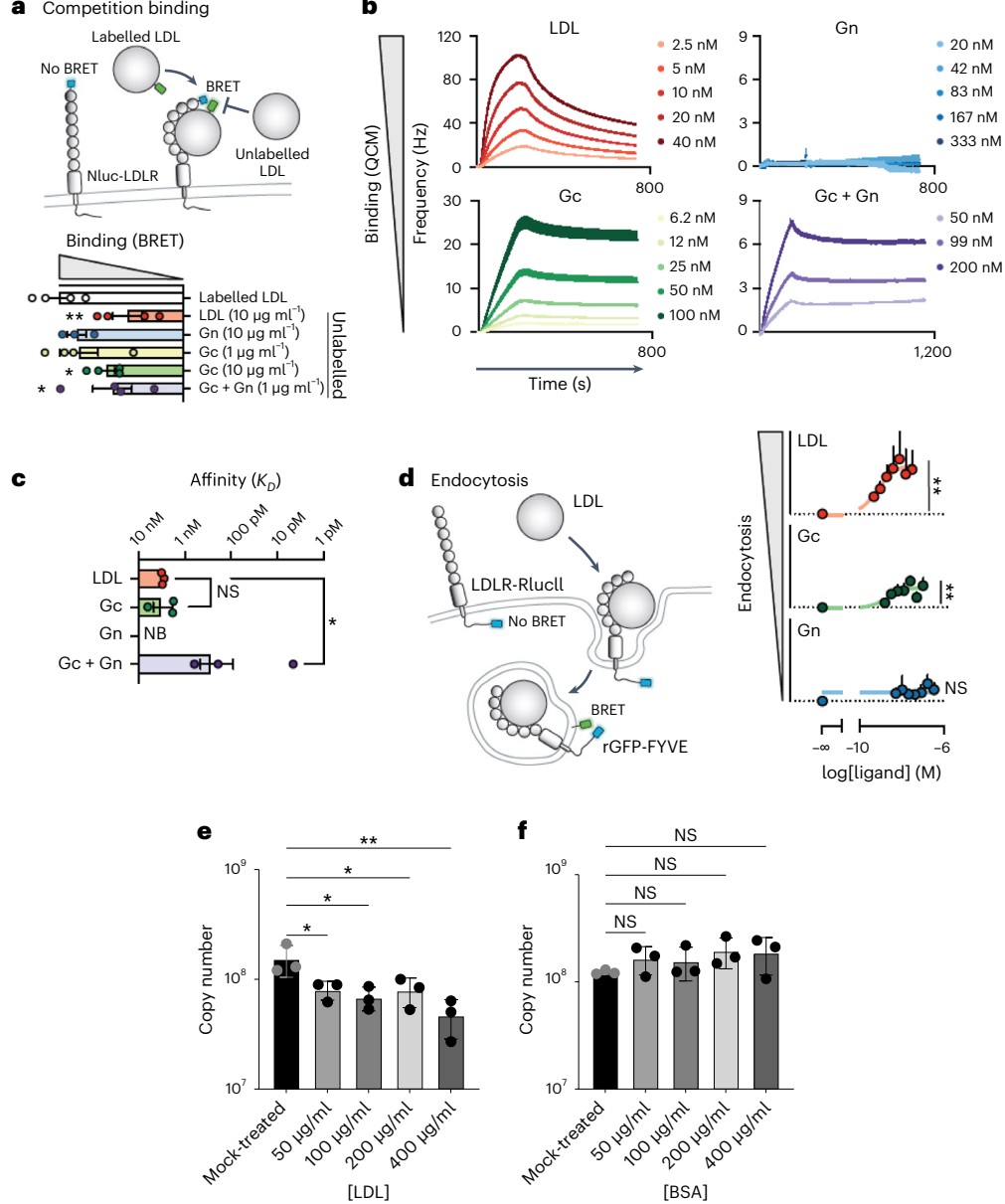

**Fig. 2 | Binding of CCHFV glycoproteins to LDLR induces receptor-mediated endocytosis. a**, Illustration depicting the BRET-based binding assay that was used to indirectly measure the binding of unlabelled ligand by outcompeting BODIPY-FL labelled LDL for interaction with Nluc-tagged LDLR. BRET between Nluc-LDLR and BODIPY-FL LDL was measured following co-administration with unlabelled LDL, CCHFV Gc, Gn or Gc/Gn, and the AUC was normalized to vehicle treatment. Data are presented as mean ± s.e.m. of $n = 4$ biologically independent experiments; *$P < 0.05$, **$P < 0.01$ (one-way ANOVA with Fisher's LSD test). **b**, Kinetic QCM experiments monitoring the interaction between LDL, CCHFV Gc, Gn or Gc/Gn with the extracellular domain of LDLR. Data are presented as mean ± s.e.m. of $n = 3$ independent experiments. **c**, Bar graph of the affinities of LDL, CCHFV Gc, Gn or Gc/Gn from QCM experiments. Data are presented as mean ± s.e.m. of $n = 3$ biologically independent experiments; NB, no binding;

*$P < 0.05$ (Kruskal–Wallis test with uncorrected Dunn's test). **d**, Schematic of the internalization assay to assess the ligand-dependent accumulation of LDLR at early endosomes. Cells expressing LDLR-RlucII (donor) and rGFP-FYVE (acceptor) were stimulated with vehicle or increasing concentrations of LDL, recombinant CCHFV Gc or recombinant CCHFV Gn for 45 min before BRET measurements. Data are presented as mean ± s.e.m. ($n = 3$ independent experiments). Binding and internalization were assessed by comparing the top and bottom parameters from nonlinear regression in the extra sum-of-squares *F*-test ($P < 0.05$). **$P < 0.01$; one-tailed extra sum-of-squares *F*-test. **e**, Competition assay between CCHFV and LDL in SW13 cells (MOI 0.01, 24 h.p.i.). **f**, BSA was used as control. Data are represented as mean ± s.d. of $n = 3$ independent experiments. *P* values were calculated using one-way ANOVA. *$P < 0.05$, **$P < 0.01$; NS $P > 0.05$. Exact *P* values are available in.

---

mice susceptible to the infection, as reported previously[42,43]. We therefore treated wild-type and *Ldlr*[−/−] mice with 2.5 mg of anti-IFNα receptor antibodies at the time of CCHFV infection (400 plaque forming units per mouse). In the initial experiment, one set of wild-type mice and one set of LDLR knockout (KO) mice were subjected to treatment and subsequently infected with CCHFV. In the subsequent experiment, one set of wild-type mice and two sets of *Ldlr* KO mice underwent treatment and CCHFV infection. At 3 days post infection (for the second experiment)

or 4 days post infection (for the first experiment), the wild-type mice reached the euthanization point (0.8 points according to the approved scoring system presented in Extended Data Table 3) with clinical signs of the disease. At this juncture, both the wild-type mice and one group of *Ldlr* KO mice were euthanized simultaneously. Serum, liver and spleen samples were collected from these mice and analysed for viral RNA. In addition, liver specimens were examined for pathologies using histology. In these cohorts, while all wild-type mice reached the final score,

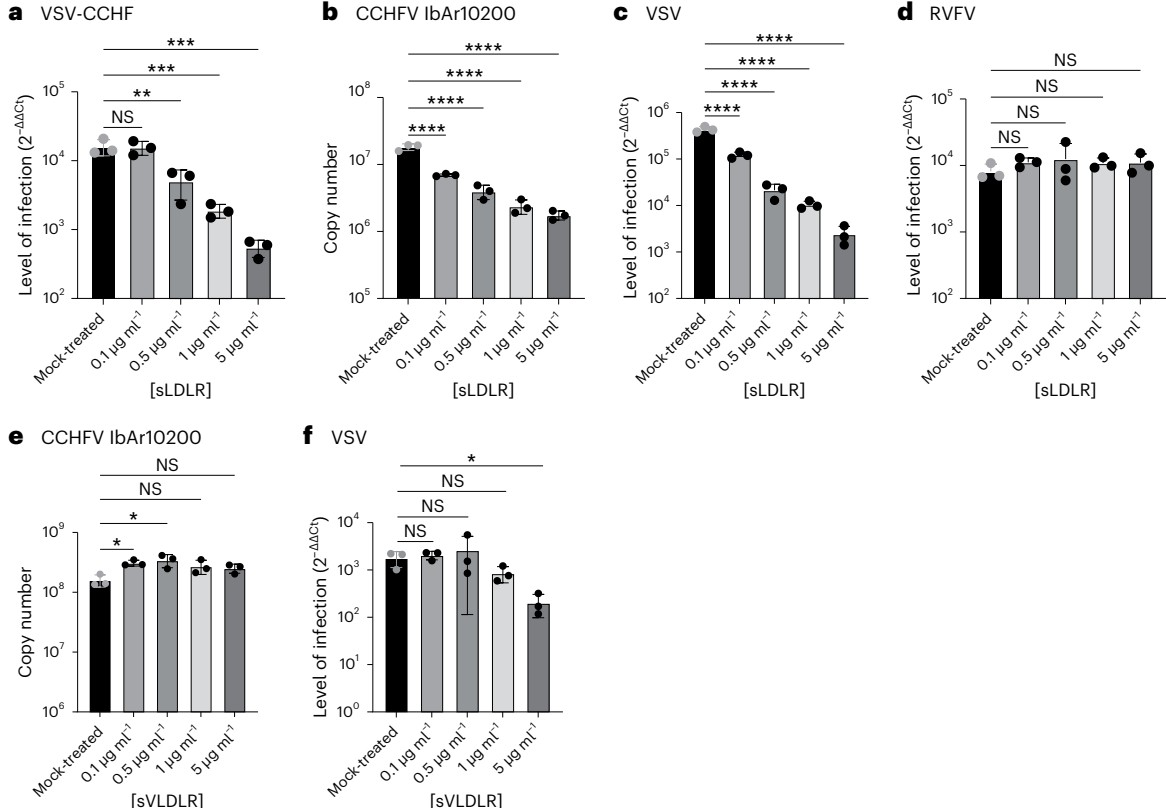

**Fig. 3 | Inhibition of CCHFV infections by soluble LDLR. a**, Levels of VSV-CCHF_G infections in human SW13 cells treated with the indicated range of soluble LDLR concentrations or left untreated (mock-treated) (MOI 0.01, 6 h.p.i.). **b**, Levels of IbAr10200 CCHFV infections in SW13 cells treated with a range of soluble LDLR concentrations (MOI 0.01, 24 h.p.i.). **c**, Levels of VSV infection in SW13 cells treated with the indicated concentrations of soluble LDLR (MOI 0.01, 6 h.p.i.).

**d**, Levels of RVFV infection of SW13 cells treated with soluble LDLR (MOI 0.01, 24 h.p.i.). **e**, Levels of IbAr10200 CCHFV infection in SW13 cells treated with soluble VLDLR decoys (MOI 0.01, 24 h.p.i.). **f**, Levels of VSV infection in SW13 cells treated with soluble VLDLR (MOI 0.01, 6 h.p.i.). Data are mean ± s.d. of $n = 3$ independent experiments. One-way ANOVA; *$P < 0.05$, **$P < 0.01$, ***$P < 0.001$, ****$P < 0.0001$, NS $P > 0.05$. Exact $P$ values are available in.

50% of $Ldlr^{-/-}$ mice displayed no weight loss or other macroscopic signs of disease. Despite no discernible difference in virus load in the serum, $Ldlr^{-/-}$ mice exhibited a significantly reduced level of viral RNA in the liver and spleen (Fig. 4c). However, half of the individuals ($ldlr^{-/-}$) manifested symptoms and were euthanized by day 3, while the remaining 50% exhibited a delayed onset of the disease (Fig. 4d). Histopathological analysis of livers of CCHFV-infected wild-type mice revealed midzonal necrosis (Fig. 4e top left), periportal coagulative necrosis (Fig. 4d top middle), as well as sporadic necrosis of single cells (Fig. 4e top right), accompanied by severe vascular congestion with low to moderate numbers of intravascular macrophages, neutrophils and occasional fibrin thrombi (Fig. 4e top middle). Interestingly, livers from 50% of the CCHFV-infected $Ldlr$ knockout mice showed little to no evidence of these pathologies (Fig. 4e bottom), indicating that $Ldlr$ knockout can, for a while, protect mice from liver damage due to CCHFV infection.

### LDLR is a receptor for CCHFV isolates

To investigate whether LDLR is a receptor for clinical isolates of CCHFV, we isolated and cultured a CCHFV from a Turkish patient sample. Consistent with the results using the laboratory strain IbAr10200, addition of sLDLR, but not sVLDLR, reduced the infection of human cells exposed to the clinical CCHFV isolate in a dose-dependent fashion (Extended Data Fig. 6a,b). In addition, multiple $LDLR$ mutant Vero and A549 cells challenged with the CCHFV patient isolate showed significantly reduced infection rates compared with their respective LDLR-expressing wild-type control cells (Extended Data Fig. 6c). These data confirm that LDLR also acts as a receptor for patient-derived CCHFV isolates.

### The role of Apolipoprotein E in CCHFV infection

While assessing the efficacy of sLDLR and sVLDLR against CCHFV infection, we extended our investigation to the third closely related member of the LDLR family, LRP8 (LDL Receptor protein 8). As depicted in Fig. 5a, sLRP8 demonstrated capability to inhibit VSV-CCHF infection, but it proved ineffective against CCHFV IbAr10200 (Fig. 5b) or a clinical isolate (Fig. 5c). We postulated that these outcomes might be attributed to the presence of Apolipoprotein E (ApoE), a ligand for both LDLR and LRP8, on the surface of VSV-CCHF. To test this hypothesis, we conducted a neutralization assay using a previously described ApoE-neutralizing antibody[44]. The ApoE antibody effectively blocked VSV-CCHF (Fig. 5d) while demonstrating no impact on CCHFV IbAr10200 (Fig. 5e).

To corroborate that the neutralization effect was indeed due to the presence of ApoE on the virus's surface, CCHFV IbAr10200 was produced using HepG2 cells, known for their high ApoE expression, and HepG2 ApoE knockout cells. As illustrated in Fig. 5f, the ApoE antibody neutralized CCHFV produced on HepG2 cells, whereas it failed to neutralize CCHFV produced on HepG2 ApoE knockout cells (Fig. 5g). These findings underscore the importance of the host cells used in culturing CCHFV in determining the composition of the progeny virus.

### LDLR and LRP8 as receptors in natural infections

To investigate the involvement of LDLR and LRP8 in natural infections, CCHFV IbAr10200 was initially cultured on a $Hyalomma$ tick cell line, and the potential inhibitory effects of sLDLR and sLPR8 were evaluated. As anticipated, sLDLR demonstrated the ability to block the virus (Fig. 6a). Interestingly, sLRP8 did not exhibit a blocking effect (Fig. 6a), highlighting the absence of ApoE on virus cultured in tick cells. Subsequently,

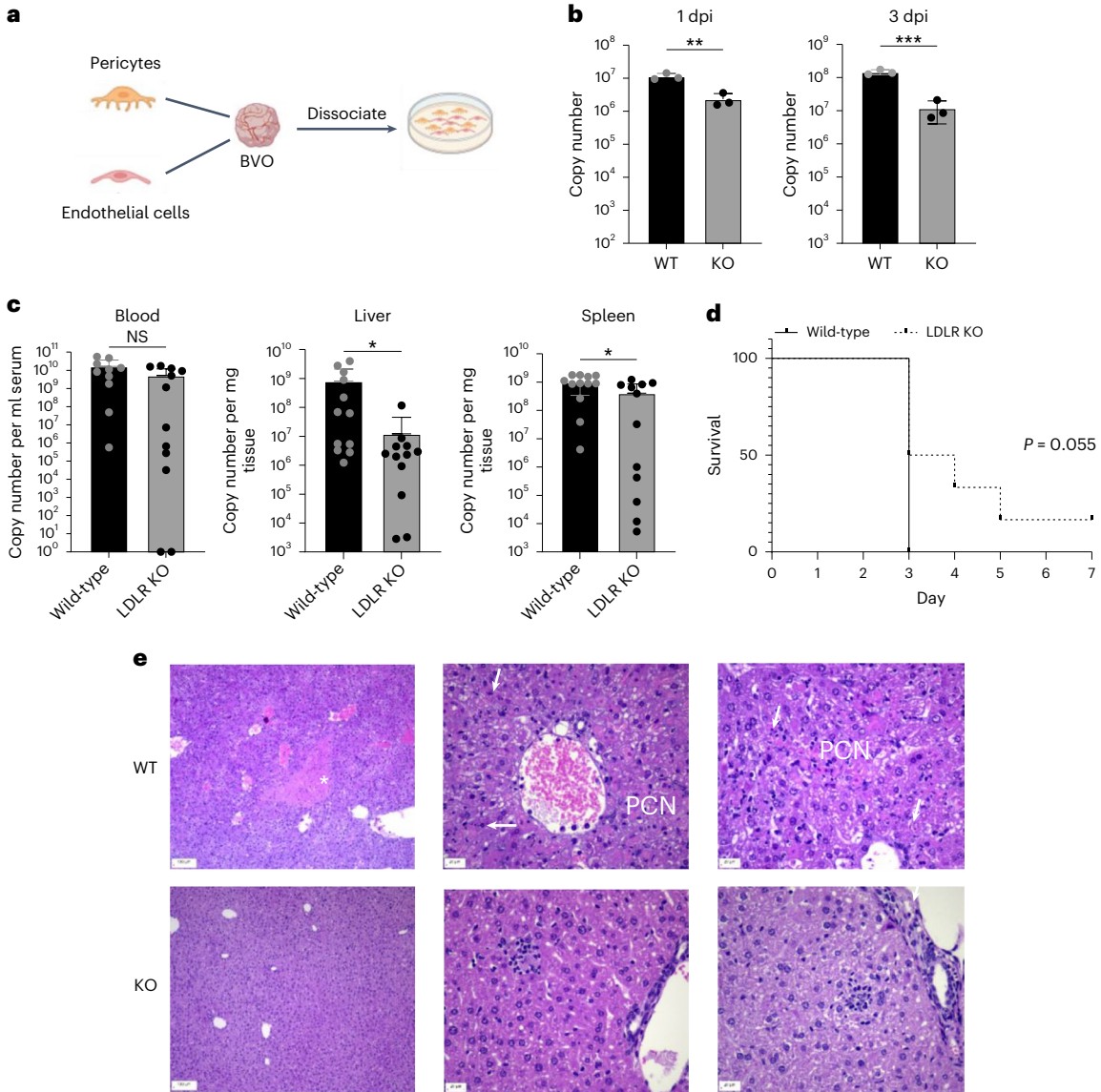

**Fig. 4 | CCHFV infections in human BVOs and *Ldlr* mutant mice. a**, Scheme representing blood vessel organoids made from *LDLR*+ and *LDLR*− iPSC cells that were dissociated and seeded as a 2D monolayer. **b**, Level of infection of CCHFV (IbAr10200) BVO-derived vascular cells generated from WT and *LDLR* KO iPSCs. Copy numbers of CCHFV RNA were determined by RT–qPCR at 1 day post infection (d.p.i.) and 3 d.p.i. (MOI 0.1). *P* values were calculated using two-sided unpaired Student's *t*-tests. *n* = 3 independent experiments. **c**, CCHFV (IbAr10200) infections of wild-type or Ldlr KO mice. *n* = 12 female mice per group (400 p.f.u.s per mouse). Numbers of CCHFV RNA copies in serum, liver and spleen of wild-type and *Ldlr* KO mice determined on the day of euthanasia. *P* values were calculated using two-sided unpaired Student's *t*-tests comparing two groups.

Data are mean ± s.d. *\*P* < 0.05, *\*\*P* < 0.01, *\*\*\*P* < 0.001. **d**, Survival of wild-type and *Ldlr* KO mice. Survival analysis was done using the Kaplan–Meier test. **e**, Histopathological analysis (H&E staining) of livers from wild-type and some *Ldlr* KO mice showing little to no pathology in the *Ldlr* KO mice infected with CCHFV and analysed on day 4 after infection. White *, midzonal necrosis; PCN, periportal coagulative necrosis; arrows, sporadic necrosis of single cells in livers of wild-type mice. Livers of most of the *Ldlr* KO mice euthanized at the same time as wild-type mice showed little to no pathology. Scale bars left: 100 μm; middle and right: 20 μm. Pictures are representative of 3 mice per group. Exact *P* values are available in.

the capacity of LDLR and LRP8 to block CCHFV was examined using virus present in patient serum. As illustrated in Fig. 6b, both sLDLR and sLRP8 (through ApoE) were effective in blocking the virus. These results underscore the importance of LDLR as a critical receptor during the transmission of the virus from ticks to humans, while both LDLR and LRP8 can be utilized by the virus throughout the course of infection in the human body. A summary of these findings is presented in Fig. 6c.

## Discussion

Understanding the molecular pathogenesis and microbe-replication cycle in host cells or host organs is essential for the development of new antivirals. CCHFV is the most widespread member of the highly lethal

haemorrhagic fever viruses with yet unknown entry receptor(s) and no current effective preventative or therapeutic options. To develop strategies to combat the increasingly alarming spread of CCHFV virus infections, we aimed to better understand the complex molecular interplay between CCHFV and host cells, especially during cell entry at the initial infection phase. Despite considerable research efforts, the CCHFV–host cell interactions remained largely elusive. This can in part be attributed to the fact that one needs to combine highest security-level infrastructures for risk class 4 pathogens (BSL-4) with cutting-edge high-throughput and genome-wide screening and validation systems. Because of the ever-increasing spread of lethal bunyaviruses and geographic expansion of their vectors to regions previously free of these viruses, we have

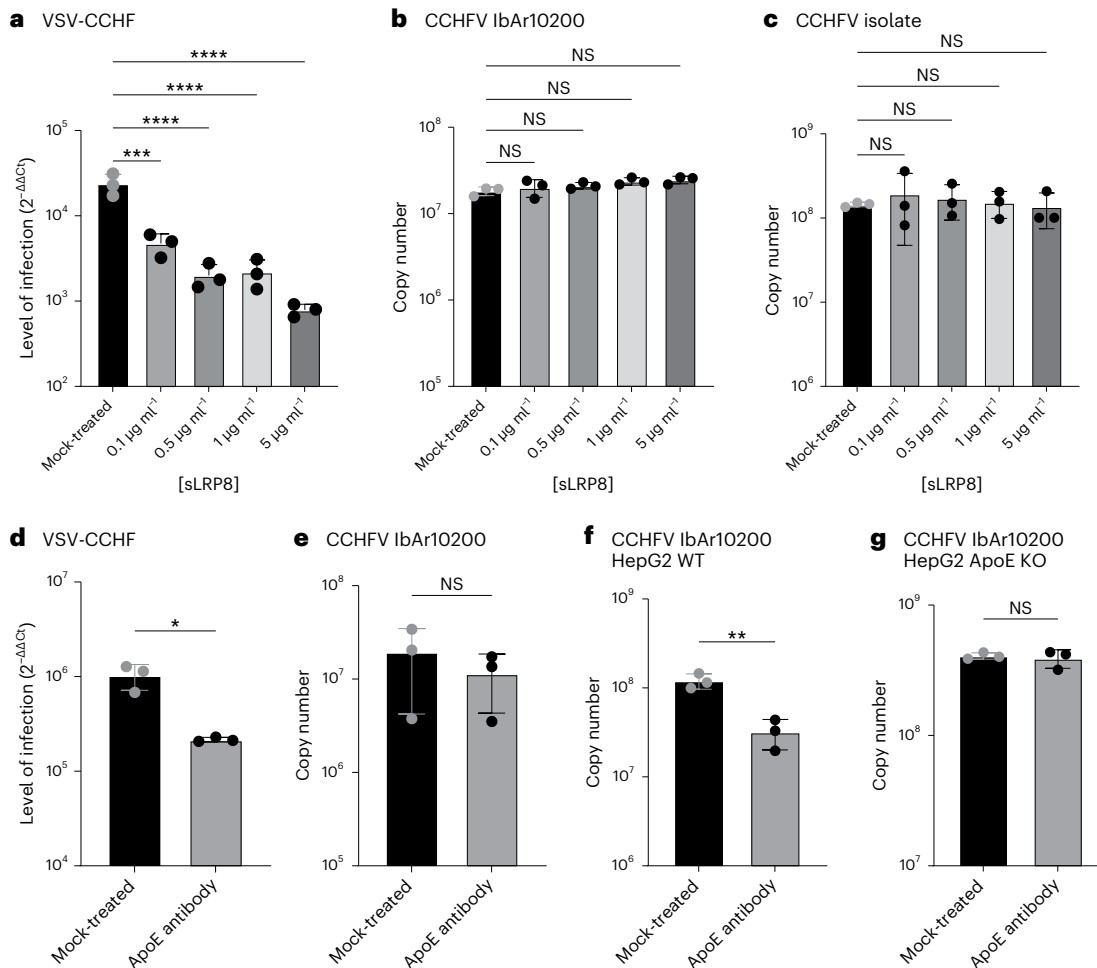

**Fig. 5 | Role of ApoE in CCHFV infection. a–c,** Level of infection in SW13 infected with VSV-CCHF (**a**), CCHFV IbAr10200 (**b**) and CCHFV isolate (**c**) treated with decoy receptor LRP8 (MOI 0.01, 6 h.p.i. or 24 h.p.i.). Data are mean ± s.d. of $n$ = 3 independent experiments. $P$ values were calculated using one-way ANOVA. **d–g,** Neutralization assay of VSV-CCHF produced on HEK293 cells (**d**), CCHFV IbAr10200 produced on SW13 cells (**e**), CCHFV produced on HepG2 cells (**f**) and CCHFV produced on HepG2 ApoE KO cells (**g**) using a neutralizing anti-ApoE antibody (MOI 0.01, 6 h.p.i. or 24 h.p.i.). Data are mean ± s.d. of $n$ = 3 independent experiments. $P$ values were calculated using two-tailed Student's $t$-test. *$P$ < 0.05, **$P$ < 0.01; ***$P$ < 0.001; ****$P$ < 0.0001. NS $P$ > 0.05. Exact $P$ values are available in.

brought together cutting-edge technologies with BLS-4 capacities for animal studies and experiments with patient isolates.

Using our haploid screening strategy in combination with a pseudotyped CCHFV, we identified LDLR as a cellular entry receptor for CCHFV. Our data show that surface Gc, and with higher-affinity Gn-Gc glycoproteins of CCHFV, directly binds to LDLR and thereby mediate virus entry via endocytosis. LDLR is a type I transmembrane protein involved in receptor-mediated endocytosis of lipoproteins, such as low-density lipoprotein (LDL) and very low-density lipoprotein (VLDL), via attachment to ApoB and ApoE present on LDL/VLDL particles. This process regulates cholesterol homoeostasis in the body[45]. Interestingly, LDLR and other members of the LDL receptor superfamily have previously been identified as key entry receptors or have been associated with cell entry for other viruses, such as vesicular stomatitis virus[28], hepatitis C virus[46,47], hepatitis B virus[48], rhinoviruses[49] and others, indicating that this broadly expressed receptor family has been repurposed for cellular entry by multiple virus clades during evolution.

Importantly, our data demonstrate that the knockout of *Ldlr* in cell lines from different species and in human blood vessel cells markedly reduced CCHFV infections. Moreover, *Ldlr* mutant mice had reduced levels of CCHFV in their liver and spleen. The knockout of *Ldlr* also led, to some extent, to the delay of the disease, providing direct in vivo evidence of the role of LDLR in the infection of CCHFV. We previously reported that CCHFV enters polarized cells predominantly from their basolateral side[50,51]. This observation is in line with earlier investigations that revealed a basolateral localization of LDLR in polarized cells[52,53]. Furthermore, we also demonstrated that exposing polarized cells to supernatants derived from CCHFV-infected dendritic cells containing cytokines, such as TNF-α and IL-6, can lead to increased CCHFV infections on the basolateral side[51]. These cytokines have been linked to severe forms of CCHF, and remarkably, they both increase the expression level of LDLR and its expression on the cell surface[54–57].

Crucially, this study underscores the importance of cell-derived proteins that may be associated with CCHFV particles, influencing the virus's ability to infect cells. Specifically, we elucidated the pivotal role of ApoE in infection through LDLR and LRP8 when the virus is generated within the human body. In contrast, CCHFV produced on tick cells predominantly utilizes only LDLR (and potentially other unidentified receptor(s)) for entry into human cells.

Notably, LDLR is recognized for its importance in the entry of hepatitis C and B viruses, despite not being a direct receptor for the glycoproteins of these viruses but through ApoE[46–48]. However, our data clearly demonstrate that CCHFV Gc-Gn directly interact with LDLR.

The exploration of cell-derived proteins present on the surface of the virus produced in human cells holds paramount importance in comprehending the intricate mechanisms underlying CCHFV infection.

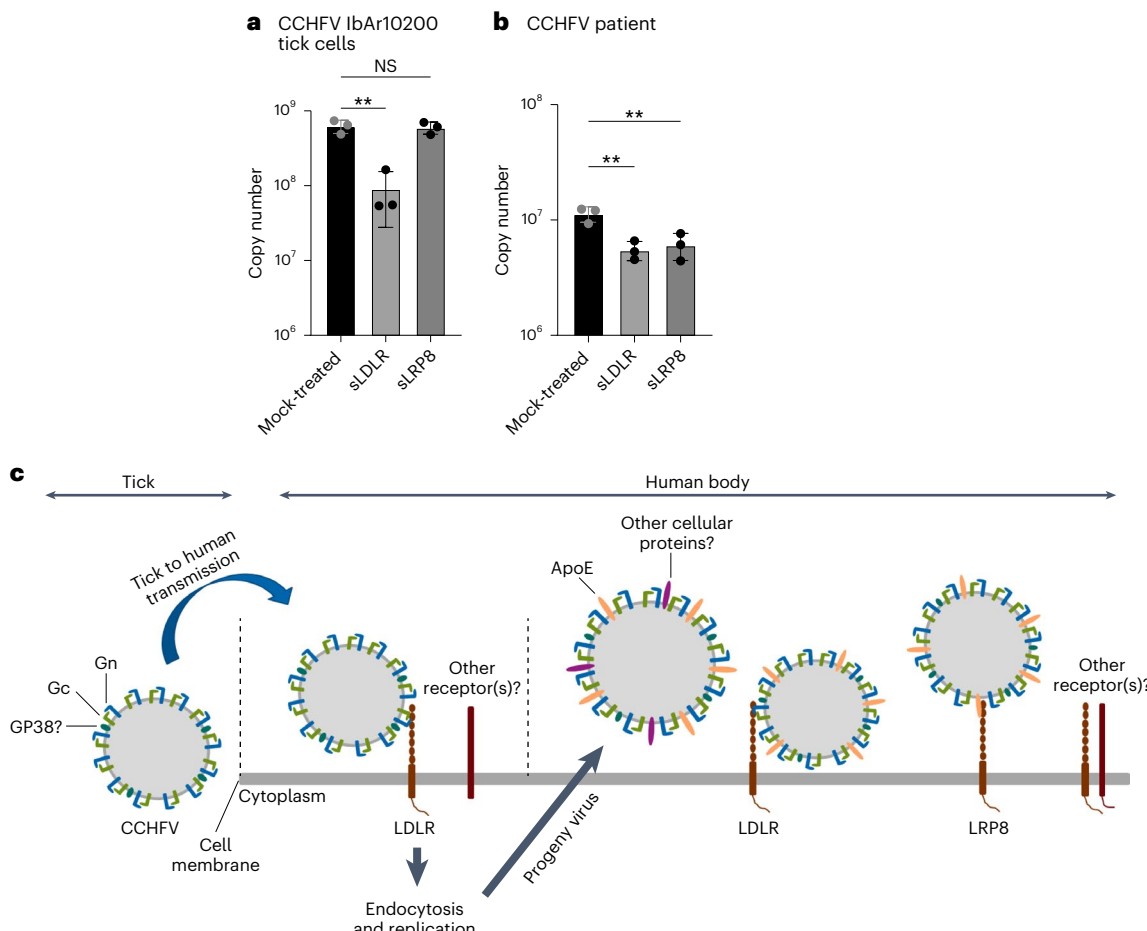

**Fig. 6 | Role of LDLR and LRP8 in the infection of CCHFV from ticks or human serum. a,b**, CCHFV produced on *Hyalomma* tick cells (**a**) and CCHFV from human patient serum (**b**) were tested for blocking with sLDLR or sLRP8 (MOI 0.01, 24 h.p.i.). Data are mean ± s.d. of *n* = 3 independent experiments. *P* values were calculated using two-tailed Student's *t*-test. **\*\****P* < 0.01; NS *P* > 0.05. **c**, Scheme representing CCHFV infection mechanisms. Exact *P* values are available in.

Recently, LDLR was proposed as a receptor for CCHFV[58]. These findings align with ours, solidifying LDLR's role as a key receptor for CCHFV. However, we also demonstrated that Gc can indeed bind to LDLR, with enhanced attachment facilitated by the synergistic influence of Gn. This sheds new light on the virus's attachment process to its receptors. In addition, our research reveals the presence of the cellular protein ApoE on the virus particle surface and its capability of attaching to both LDLR and LPR8. This dual attachment mechanism contributes to a more efficient entry of CCHFV into cells.

CCHFV is listed as a key priority in WHO's list of infectious agents with epidemic or pandemic potential. Recent outbreaks in ever-expanding geographic regions due to the effects of climate change make it paramount to identify critical components of CCHFR infections. Our study identifies a key receptor involved in CCHFV cell entry and infection. Importantly, we also show that soluble LDLR decoys can effectively reduce CCHFV infections, regardless of whether they are produced by ticks or in the human body. Such decoys can be rapidly developed as a much-needed antiviral strategy to prevent and/or treat endemic and epidemic infections with the highly lethal haemorrhagic fever virus CCHFV.

## Methods

Our research complies with all relevant ethics regulations of Karolinska Institutet and the Public Health Agency of Sweden. The animals were housed according to Karolinska Institute ethics rules and observed daily. The Stockholm Ethical Committee for animal research approved the research. Ethics clearance for patient sampling was approved by the Turkish Ethical Committee and the Bulgarian Ethical Committee. All volunteers gave written informed consent. The use of these samples for research in Sweden was approved by the Stockholm Regional Ethical Committee (2017/1712-31/2).

### Cells and viruses

The cell lines used were HEK293 (ATCC, CRL-1573), HEK293T/17 (HEK293T, ATCC CRL-11268), A549 (ATCC CCL-185), HepG2 (Abcam, AB275467), HepG2 ApoE KO (Abcam, AB280875) and Vero cells (ATCC CCL-81). All cell lines were maintained in Dulbecco's modified eagle's medium (DMEM, Life Technologies), supplemented with 10% v/v of heat-inactivated fetal bovine serum (FBS, Life Technologies) and incubated at 37 °C, 95% humidity and 5% $CO_2$. SW13 (ATCC, CCL-105) cells were maintained in Leibovitz's L15 medium (ThermoFisher) at 37 °C without $CO_2$. Haploid mouse stem cells (mSCs, clone AN3-12) used for the haploid screening were obtained from IMBA (Austria). Haploid mSCs were maintained in standard embryonic stem-cell medium, supplemented with 10% (v/v) FBS (Hyclone), recombinant mouse Leukaemia Inhibitory Factor (LIF) and β-mercaptoethanol at 37 °C, 95% humidity and 5% $CO_2$. *Ldlr* knockout AN3-12 cells were furnished (and validated) by Haplobank[27], IMBA, Vienna. The *Hyalomma anatolicum* embryo-derived cell lines HAE/CTVM9 were grown in L15/MEM medium (equal volumes of L15 and minimal essential medium with Hank's salts supplemented with 10% Tryptose Phosphate Broth), both supplemented with 2 mM l-glutamine, 20% FBS and incubated in sealed flasks at 28 °C and 0% $CO_2$ as previously described[59]. All cell lines were regularly tested for mycoplasma contamination.

VSV-CCHF_G was produced as described below. CCHFV IbAr10200 strain was cultured on SW13 cells. CCHFV clinical strain was isolated on SW13 cells from a Turkish patient serum sampled as part of another project. Ethics clearance was obtained (Nr: 2017/1712-31/2) as well as fully informed patient consent. RVFV strain ZH548 was cultured on Vero cells.

## Biosafety

All experiments involving VSV-CCH_G were done in a Biosafety Level 2 laboratory and experiments involving CCHFV were done in a Biosafety Level 4 laboratory in compliance with the Swedish Public Health Agency guidelines (Folkhälsomyndigheten, Stockholm).

## Reagents

D-PBS, DMEM, trypsin, PBS, penicillin/streptomycin and FBS were from Gibco (ThermoFisher). Polyethylenimine (PEI) was purchased from Alfa Aesar (ThermoFisher). Unlabelled LDL from human plasma and BOPIDY FL complexed LDL were purchased from ThermoFisher. Human Fc-tagged CCHFV Gc, 6×His-tagged CCHFV Gn and BODIPY-FL complexed CCHFV Gc were purchased from Native Antigen. Coelenterazine h was purchased from Nanolight Technologies. NanoBRET Nano-Glo substrate was purchased from Promega. Trizol was purchased from ThermoFisher. Anti-IFN type I receptor antibody (MAR1-5A3) was purchased from Leico (MAR1-5A3 [5A3]; Leinco Technologies). Soluble LDLR, VLDLR and LRP8 were purchased from R&D Systems.

## Pseudotyped virus production and titration

The plasmid pC-G[7] expressing the CCHFV glycoproteins Gn and Gc (strain IbAr10200) was kindly provided by Robert A. Davey (Texas Biomedical Research Institute, San Antonio, Texas, USA). The plasmid expressing VSV glycoprotein (pVSV-G) was previously described[60]. The recombinant VSV encoding the GFP in place of the VSV-G gene (VSVΔG-GFP) was kindly provided by Michael Whitt (University of Tennessee, USA). CCHFV-Gn/Gc-pseudotyped VSVΔG-GFP (CCHFV-pseudotyped virus) was generated as previously described[7]. Briefly, HEK293T cells were seeded in a T75 flask and 24 h later transfected using the calcium-phosphate protocol with 20 μg of pC-G plasmid; 24 h later, the cells were infected with the recombinant VSVΔG-GFP virus at a multiplicity of infection (MOI) of 4 fluorescent focus-forming units (f.f.u.) per cell. At 16 h.p.i., cell culture supernatants were collected and cell debris were cleared by centrifugation (1,200 $g$ for 7 min at 4 °C). Thereafter, virus particles were pelleted by ultracentrifugation (300,000 $g$ for 150 min at 4 °C) on a 20% (p/v) sucrose cushion in a Beckmann SW 28 Ti swinging-bucket rotor. Pellets were resuspended in 1 ml of ice-cold 1X PBS per tube and mixed. Subsequently, the virus was aliquoted and stored at −80 °C until use. Virus titre was determined by immunofluorescence on Vero cells seeded on 96-well plates. Viral stock was 10-fold serially diluted in DMEM and inoculated on confluent Vero cells for 1 h at 37 °C. Cells were then washed and DMEM supplemented with 10% FBS was added. After 18 h, cells were fixed in chilled methanol/acetone and stained with VSV-M protein (VSV-M [23H12], Kerafast) antibody Alexa Fluor 488-conjugated goat anti-mouse IgG secondary antibodies (ThermoFisher). The fluorescent foci were counted and viral titre was expressed as f.f.u. ml[−1]. To confirm the functionality of the glycoprotein complex in our experimental conditions, we ran a seroneutralization test with serum from a vaccinated Bulgarian lab worker and with control (unvaccinated people) sera.

## Chemical mutagenesis of haploid stem cells

Chemical mutagenesis using ENU was performed as described previously. Briefly, haploid AN3-12 cells were treated for 2 h with 0.1 mg ml[−1] ENU in full medium while in suspension and under constant agitation. Cells were washed 5 times and transferred to a culture dish. Cells were left to recover for 48 h, separated using trypsin/EDTA and frozen in 10% DMSO, 40% FBS and 50% full medium. ENU libraries as well as untreated control libraries were shipped to Stockholm for screening experiments using VSV-CCHFV.

## Haploid cell screens and analysis

Haploid mSCs (50 million) were thawed and infected with VSV-CCHF_G at a high MOI of 10 (to enhance the likelihood of infecting all susceptible cells) in 5 ml of ES medium without FBS. At 1 h after infection, the cells were supplemented with complete ES medium and incubated at 37 °C with 5% $CO_2$. After outgrowth of virus-resistant cells, cell clones were picked separately and cultured before being validated by infection assay with CCHFV IbAr10200. Briefly, cells (AN3-12 wild-type and potentially resistant clones) were seeded at $5.0 \times 10^4$ cells per well in DMEM and 5% FBS for 24 h. They were then infected with CCHFV at an MOI of 0.1, the cells recovered 24 h post infection in Trizol and then analysed by RT–qPCR. All clones that were fully or partly resistant to CCHFV infection were subjected to DNA extraction using the Gentra Puregene tissue kit (Qiagen). Paired-end 150-bp whole-exome sequencing was performed on an Illumina Novaseq 6000 instrument after precapture barcoding and exome capture with the Agilent SureSelect Mouse All Exon kit. For data analysis, raw reads were aligned to the reference genome mm9. Variants were identified and annotated using GATK 4.5.0.0 and snpEff 5.2. CCHFV resistance causing alterations were identified by allelism, only considering variants with moderate or high effect on protein and a read coverage >20.

## Generation of LDLR knockout cells

A549 (ATCC, CCL-185) and Vero (CCL-81) cells were grown in complete DMEM medium (DMEM high glucose supplemented with 10% FBS (Gibco), 1x MEM-NEAA (Gibco), 1x glutamax (Gibco), 1 mM sodium pyruvate (Gibco) and 100 U ml[−1] penicillin-streptomycin (Gibco)). The day before transfection, $1.05 \times 10^5$ cells were seeded per well of a 24-well plate in 0.5 complete DMEM medium. The next day, the culture medium was replaced with fresh complete DMEM medium and transfected with a liposome:DNA mixture composed of 50 μl Opti-MEM I (Gibco), 500 ng of PX459 v2.0 plasmid (Addgene 62988, Puro resistant), 1.5 μl Lipofectamine 3000 reagent and 1.0 μl P3000 reagent. Several single guide RNAs were derived from CRISPick (https://portals.broadinstitute.org/gppx/crispick/public) using SpCas9 Cas9 knockout and the human LDLR gene as input. The final guide RNA sequence used for knockout studies was gATGAACAGGATCCACCACGA (lower letter g denotes preceding guanosine to enhance transcription from the U6 Promoter). The next day, the medium was replaced with complete DMEM supplemented with 1 μg ml[−1] puromycin for transient selection. At 60 h post transfection, each well containing selected A549 or Vero cells were expanded to 1 well of a 6-well plate in complete DMEM medium. Once cells reached 80% confluency, they were dissociated with 500 μl TrypLE Express enzyme solution (Gibco) for 5 min and collected in FACS buffer (D-PBS containing 5% FBS). After one wash with FACS buffer, 10 μl of α-LDLR-PE antibody (R&D Systems, FAB2148P) per $1.0 \times 10^6$ cells were added and stained for 1 h on ice in the dark. Unmodified cells were used as controls. After 1 h of staining, cells were collected by centrifugation and washed twice in FACS buffer. Finally, cells were resuspended in 1 ml of FACS buffer and LDLR-negative cells were sorted into individual wells of a 96-well plate. LDLR-negative cells were defined as single cells displaying no PE fluorescence. Individual clones were expanded and analysed. Data were analysed during sorting with BD FACSDiva (v.9.0.1) and re-analysed for plotting of data presented in this manuscript using FlowJo (10.8.1). Unmodified A549 or Vero cells, as well as bat Tb-1 Lu cells (ATCC, CCL-88) were used as positive and negative controls, respectively. When individual cells grew to 85% confluency, they were expanded onto 24-well plates. After expansion, LDLR gene editing was verified by flow cytometry analysis using the α-LDLR-PE antibody as described above and genotyped using the forward primer F: CTAACCAGTTCCTGAAGC and reverse primer R: GCACCCAGCTTGACAGAG. For genotyping, $5.0 \times 10^4$ cells

were collected and resuspended in 100 µl of nuclease-free water. DNA QuickExtract lysis solution (100 µl, Lucigen) was added and incubated for 5 min at 65 °C and 5 min at 95 °C. Of the lysis solution, 2 µl were used per 20 µl of PCR reaction containing 1x Kapa HiFi HotStart ReadyMix (Roche) and 0.5 µM of each forward and reverse primer. PCR was performed with an initial 3-min 95 °C denaturation step, followed by 35 cycles of 98 °C for 10 s, annealing at 58 °C for 20 s, extension for 1 min at 72 °C and a final extension for 2 min at 72 °C. PCR products were purified and subjected to Sanger sequencing for verification. Cells that showed Cas9 editing at the LDLR locus and negative α-LDLR staining were used as knockout for entry studies.

### Cell infection
For all infections involving AN3-12, A549 and Vero cells, $5.0 \times 10^4$ cells per well were seeded in 48-well plates (Sarstedt). At 24 h post seeding, cells were infected with either VSV, VSV-CCHF_G, CCHFV (IbAr10200 or isolate) or RVFV at an MOI of 0.1 for 1 h in corresponding media containing 2% FBS. After 1 h, cells were washed once with PBS, and fresh medium containing 5% FBS was added. At 24 h (A549 and Vero) or 48 h (AN3-12) post infection, cells were washed three times with PBS and lysed with Trizol. RNA was extracted and analysed by RT–qPCR as described below.

### Soluble LDLR, VLDLR and LRP8 assays
SW13 were seeded at a density of $5.0 \times 10^4$ cells per well in a 48-well plate. At 24 h post seeding, cells were counted to define the quantity of virus needed for an infection at an MOI of 0.01. The virus was then mixed in 1.5 ml tubes (Sarstedt) with the appropriate quantity of sLDLR (R&D systems), sVLDR (R&D systems) or sLRP8 (R&D systems) in L15 medium containing 0.5% FBS. The tubes were then incubated for 30 min under shaking (75 r.p.m.) at 37 °C. After 30 min, cells were rinsed once with PBS before being infected with virus only or with the mix virus/ sLDLR, virus/sLRP8 or virus/VLDLR for 1 h at 37 °C. After 1 h, inocula were removed, cells washed once with PBS and L15 medium containing 5% FBS added to each well. VSV and VSV-CCHF_G entering cells and replicating very fast, cells infected with these viruses were recovered at 6 h post infection, while cells infected with CCHFV and RVFV were recovered at 24 h post infection. At the time of recovery, cells were washed three times with PBS and lysed with Trizol. RNA was extracted and analysed by RT–qPCR as described below.

### Plasmid DNA constructs for BRET assay
To generate LDLR-RlucII, codon-optimized LDLR was synthesized as a gBlock (Integrated DNA Technologies) and subcloned by Gibson assembly in pcDNA3.1/Hygro(+) GFP[10]-RlucII db v.2 that had been linearized by PCR to exclude GFP[10]. To generate Nluc-LDLR, codon-optimized LDLR from LDLR-RlucII was amplified by PCR and subcloned by Gibson assembly in pcDNA3.1 Nluc-synFZD_5 that had been linearized by PCR to exclude FZD_5. rGFP-FYVE has been described previously[33]. All plasmid constructs were verified by Sanger sequencing.

### Cell culture and transfection for BRET assay
HEK293 cells were propagated in plastic flasks and grown at 37 °C in 5% $CO_2$ and 90% humidity. Cells (350,000 in 1 ml) were transfected in suspension with 1.0 µg of plasmid DNA complexed with linear PEI (MW 25,000, 3:1 PEI:DNA ratio).

### BRET assays
**Receptor trafficking.** To monitor the trafficking of LDLR to early endosomes, HEK293 cells were transfected with LDLR-RlucII and rGFP-FYVE, and seeded in 6-well plates ($7.0 \times 10^5$ cells per well). After a 48-h incubation, cells were washed once with HBSS solution, detached and resuspended in HBSS containing 0.1% BSA, distributed into white 96-well plates containing serial dilutions of LDL, CCHFV Gc, CCHFV Gn or SARS-CoV-2 RBD, and returned to the incubator for

45 min at 37 °C. Before BRET measurements, cells were incubated with coelenterazine h (10 min).

**NanoBRET binding assay.** To monitor the binding of fluorescent ligands to LDLR, HEK293 cells were transfected with Nluc-LDLR and seeded in white 96-well plates ($3.5 \times 10^4$ cells per well). After a 48-h incubation, cells were washed once with HBSS and maintained in the same buffer. Before BRET measurements, cells were incubated with NanoBRET Nano-Glo substrate (6 min) and then stimulated with either BODIPY-FL LDL or BODIPY-FL Gc for 90 min following a baseline measurement of 3 cycles. For the competition binding assay, BODIPY-FL LDL (3.75 µg ml$^{-1}$) was added together with unlabelled LDL, CCHFV Gc, CCHFV Gn or CCHFV Gc and Gn to cells expressing Nluc-LDLR for 15 min, and the area under the curve (AUC) was normalized to vehicle-treated cells.

**BRET measurements.** Plates were read on a Tecan Spark multimode microplate reader equipped with a double monochromator system to measure the emission of the RlucII/rGFP donor–acceptor pair in receptor trafficking experiments (430–485 nm (donor) and 505–590 nm (acceptor)) or the Nluc/BODIPY-FL donor–acceptor pair in the NanoBRET binding assay (445–470 nm (donor) and 520–575 nm (acceptor)).

### Quartz crystal microbalance (kinetic experiments)
The Attana cell A250 was employed for real-time binding kinetics analysis. A recombinant LDLR protein was covalently immobilized onto the Attana LNB Carboxyl Sensor Chip (3623-3103) at the specified ligand density (20 µg) using the Amine Coupling kit (3501-3001, Attana) following manufacturer recommendations. The binding of analytes (LDL as a positive control, HFVGC, HFVGN, G38, Toscana G2) occurred at 22 °C, employing a continuous flow of D-PBS with $Ca^{2+}/Mg^{2+}$ (0.3% BSA, pH 7.4) as the running buffer at a flow rate of 10 µl min$^{-1}$. Before each measurement, a reference injection (blank) of the running buffer was conducted and subtracted from the binding curves during data analysis. Sensor chips were regenerated after each measurement by injecting 10 mM glycine, pH 1.0. Consistent binding curves were observed upon repeated injections of the same analyte concentration, indicating that regeneration did not impact the surface's binding capacity. The frequency change in sensor surface resonance ($\Delta F$) during the binding experiments was recorded using the Attester software (Attana AB). The data were assessed and analysed using the Evaluation (Attana AB) and TraceDrawer software 1.9.1 (Ridgeview Instruments), employing 1:1 or 1:2 binding models to calculate kinetic parameters, including rate constants ($k_a$, $k_d$), dissociation equilibrium constant ($K_D$) and maximum binding capacity ($B_{max}$).

### LDL competition assays
SW13 were seeded at a density of $5.0 \times 10^4$ cells per well in a 48-well plate. At 24 h post seeding, cells were counted to determine the quantity of virus needed for infection at an MOI of 0.01. CCHFV was then mixed in 1.5 ml tubes (Sarstedt) with different concentration of LDL (Thermofisher, L3486) or BSA (Saveen & Werner, A1391) in L15 medium containing 0.5% FBS. Cells were rinsed once with PBS before being infected with virus only or with the mix virus/LDL or virus/BSA for 1 h at 37 °C. After 1 h, inocula were removed, cells washed once with PBS and L15 medium containing 5% FBS added to each well. Cells were recovered at 24 h post infection. At the time of recovery, cells were washed three times with PBS and lysed with Trizol. RNA was extracted and analysed by RT–qPCR as described below.

### Generation of LDLR knockout iPSC
NC8 iPSCs (male, pericyte derived) were grown on Matrigel (human embryonic stem-cells qualified, Corning) coated dishes in complete Stemflex medium (Gibco) + 1:100 antibiotic-antimycotic (Gibco) (Invivogen). Cells were passaged using 0.5 mM EDTA at a ratio of 1:6

every 3 to 4 days. The day before transfection, iPSCs were dissociated into single cells using TrypLE select (Gibco) and seeded at $5.0 \times 10^4$ cells per well of an rhLaminin521 (Gibco) coated 24-well plate in complete Stemflex medium supplemented with 1:100 RevitaCell (Gibco). The next day, the culture medium was replaced with Opti-MEM I (Gibco) + 1:00 RevitaCell and transfected with a liposome:DNA mixture composed of 50 µl Opti-MEM (Gibco), 500 ng of PX459 v2.0 plasmid with LDLR guide sequence gATGAACAGGATCCACCACGA cloned in (Addgene, 62988, Puro resistant), 1.5 µl Lipofectamine 3000 reagent and 1 µl P3000 reagent. After 4 h, the transfection mixture was removed and fresh complete Stemflex medium was added. After 48 h post transfection, complete Stemflex medium with 0.5 µg ml⁻¹ puromycin was added for transient selection. At 60 h post transfection, selection medium was removed and cells were expanded to 1 well of a 6-well plate. Once cells reached 85% confluency, iPSCs were dissociated into single cells using TrypLE select enzyme (Gibco) and resuspended in iPSC FACS buffer (D-PBS + 1% KOSR + 1:100 RevitaCell+0.5 mM EDTA). Anti-LDLR staining was done as described for A549. LDLR-negative as well as LDLR-positive cells were sorted into rhLaminin521-coated 96-well plates containing 150 µl of complete Stemflex medium + 1:100 RevitaCell. At 4 days post sorting, the medium was replaced with complete Stemflex medium until cells reached confluency. Individual clones were expanded and analysed as described for A549 cells.

### Preparation of blood vessel organoid-derived 2D monolayer for infection

Blood vessel organoids from NC8 clone 10 (LDLR+) and clone 4 (LDLR−) were produced as previously described[61]. To prepare the BVOs for infections, they were cut out of the matrix on day 11 of the procedure and cultured in sprouting media (StemPro-34 SFM medium (Gibco), 1X StemPro-34 nutrient supplement (Gibco), 0.5 ml glutamax (Gibco), 15% FCS, 100 ng ml⁻¹ VEGF-A (Peprotech) and 100 ng ml⁻¹ FGF-2 (Miltenyi Biotec)) for 5 additional days with media changes every other day. To dissociate the organoids, 25 mature blood vessel organoids per genotype were washed twice with PBS and transferred into a prefiltered and prewarmed enzymatic dissociation mix consisting of 4 mg Liberase TH (Sigma Aldrich) and 30 mg Dispase II (Life Technologies) dissolved in 10 ml PBS. The organoid containing the enzymatic mix was incubated for 25 min at 37 °C, followed by trituration 15 times with a 10 ml stripette. The 37 °C incubation and trituration were repeated for 10 min twice more. The dissociated organoids were passed through a 70 µm cell strainer into 5 ml of ice-cold DMEM/F12 medium. Following filtering, the cells were collected by centrifugation ($300 \times g$, 5 min) and replated in PureCol (Advanced BioMatrix, 30 µg ml⁻¹ in PBS for 1 h at r.t.) coated T-25 flasks at 30,840 cells cm⁻² in sprouting media.

### Ethics statement

In the current studies, we used 12 female C57BL/6J mice (000664, Charles River) and 18 female B6.129S7Ldlrtm1Her/J (*Ldlr* KO) mice (002207, Jackson Laboratory)[62]. All mice were 10 weeks old at the time of infection. The animals were housed according to Karolinska Institute ethics rules and observed daily. The Stockholm Ethical Committee for animal research approved the research. Animals were assigned to experimental groups according to their genetic backgrounds.

### Antibody treatment and challenge

To make the mice susceptible to CCHFV infection, all animals received an intraperitoneal injection of 2.5 mg anti-IFN type I receptor antibody at the time of infection[63]. Each mouse was challenged with 400 f.f.u.s of CCHFV IbAr10200 in 100 µl via intraperitoneal injection. The mice were monitored daily for clinical signs of disease and their overall well-being. When the wild-type mice reached the predetermined humane endpoint, wild-type and one group of *Ldlr* ⁻/⁻ mice were euthanized independent of clinical signs. Blood was collected in microcontainer

tubes for serum separation and serum was inactivated with Trizol for subsequent RT–qPCR analysis. In addition, liver, spleen and kidney were collected, with a portion kept in Trizol for RT–qPCR and another portion fixed in 4% paraformaldehyde for histopathological analyses. The third group of *Ldlr* ⁻/⁻ mice was monitored daily for survival and when the mice reached the predetermined human endpoint or the end of the experiment, they were euthanized.

The experimenters were not blinded to the identity of the animals. However, the pathologist who analysed livers as well as the scientist who ran the RT–qPCRs and the subsequent analysis were blinded.

### Histopathology

Paraformaldehyde-fixed livers were cut into 3–4-µm-thin sections and stained for haematoxylin and eosin (H&E). The stained sections were analysed by a pathologist at BioVet, a laboratory of animal medicine (Sollentuna, Sweden).

### ApoE neutralization assays

SW13 were seeded at a density of $5.0 \times 10^4$ cells per well in a 48-well plate. At 24 h post seeding, cells were counted to determine the quantity of virus needed for infection at an MOI of 0.01. The virus was then mixed in 1.5 ml tubes (Sarstedt) with 1:20 dilution of ApoE antibody (Sigma, AB947) in L15 medium containing 0.5% FBS. The tubes were then incubated for 30 min under shaking (75 r.p.m.) at 37 °C. After 30 min, cells were rinsed once with PBS before being infected with virus only or with the mix virus/ApoE antibody for 1 h at 37 °C. After 1 h, inocula were removed, cells washed once with PBS and L15 medium containing 5% FBS added to each well. VSV-CCHF_G entered cells and replicated very fast; cells infected with this virus were recovered at 6 h post infection, while cells infected with CCHFV were recovered at 24 h post infection. At the time of recovery, cells were washed three times with PBS and lysed with Trizol. RNA was extracted and analysed by RT–qPCR as described below.

### RT–qPCR analysis

All RNA extractions were performed using Direct-zol RNA extraction kit (Zymo Research). Quantitative real-time PCR reactions were performed using a *Taq*Man Fast Virus 1-step master mix (ThermoFisher) and run on an Applied Biosystems machine. The following primers were used in this study to detect CCHFV L gene (Fwd: GCCAACTGTGACKGTKTTCTAY-ATGCT, Rev1: CGGAAAGCCTATAAAACCTACC TTC, Rev2: CGGAAAGC-CTATAAAACCTGCCYTC, Rev3: CGGAA AGCCTAAAAAATCTGCCTTC, probe: FAM-CTGACAAGYTCAGCAAC-MGB); RVFV (Fwd: AAAATTC-CTGAGAC ACATGGCAT, Rev: TCCACTTCCTTGCATCATCTGAT, Probe: FAM-CAATGTAA GGGGCCTGTGTGGACTTGTG-TAMRA); VSV-M gene (Fwd: TGATACAGTACAATTA TTTTTGGGAC, Rev: GAGACTTTCTGT-TACGGGATCTGG, Probe: FAM-ATGATGCA TGATCCAGC-MGB). RNase P RNA was used as an endogenous control for normalization (Fwd: AGATTTGGACCTGCGAGCG, Rev: GAGCGGCTGTCTCCACAAGT, Probe: FAM-TTCTGACCTGAAGGCTCTGCGCG-MGB).

Absolute quantification of CCHFV RNA for mice samples was performed by RT–qPCR. A 120 bp synthetic RNA corresponding to nucleotides 9,625–9,744 of CCHFV Ibar 10200 L segment (GenBank MH483989.1) was produced by Integrated DNA Technologies. The standard synthetic RNA was solubilized in RNase-free water and the copy number calculated after quantification by nanodrop. The efficiency and linearity of the RT–qPCR reaction (using the primers: forward GCCAACTGTGACKGTK-TTCTAYATGCT and reverse: CGGAAAGCCTAAAAAATCTGCCTTC, with probe FAM-CTGACAAGYTCAGCAAC-MGB) with the standard RNA was validated over serial 10-fold dilutions. This standard curve RT–qPCR was then performed simultaneously with RNA samples to quantify the absolute copy number of CCHFV RNA.

### Statistical analyses

All analyses were done using the data from at least three independent experiments and are shown as mean ± s.d. in GraphPad Prism (v.9.4.1).

One-way analysis of variance (ANOVA) (with multiple comparisons Dunnett corrections) and two-tailed Student's *t*-test were used as indicated in figure legends. Data distribution was assumed to be normal, but this was not formally tested. No statistical methods were used to predetermine sample sizes but our sample sizes are similar to those reported in previous publications[34–36,51].

Data collection was not performed blind to the conditions of the experiments, but analysis was blinded.

No animals or data points were excluded from the analyses.

### Reporting summary

Further information on research design is available in the Nature Portfolio Reporting Summary linked to this article.

## Data availability

Sequencing data are available on the NCBI Sequence Read Archive under the accession number PRJNA1085501. Source data are provided with this paper.

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

## Acknowledgements

We thank M. Boehm (NIH) for providing NC8 cells for blood vessel organoids development, R. A. Davey (Texas Biomedical Research Institute, San Antonio, Texas, USA) for providing the pC-G plasmid and M. Witt (University of Tennessee, USA) for providing VSVΔG-GFP. This research was funded by Karolinska Institutet Research Foundation, grant number FS-2022:0010 (V.M.M.). The work was supported by the Swedish Research Council 2018-05766 (A.M., J.M.P. and F.W.), the Swedish Research Council (grant numbers: 2019-01837 and 2021-02801), the European Union's Horizon 2020 research and innovation program under grant agreement number 732732 (A.M., F.W.), the Ruth och Richard Julins Foundation for Gastroenterology (grant number 2021-00158) and the Knut and Alice Wallenberg Foundation (Grant VC-2021-0026) (V.M.L.). This work was also supported by the European Research Council (ERC) under the European Union's Horizon 2020 research and innovation Programme CoG-2020_101002478_ENGINORG (N.M.) and the Innovative Medicines Initiative 2 Joint Undertaking (JU) under grant agreement no. 101005026. JU was supported by the European Union's Horizon 2020 research and the innovation programme and EFPIA (A.M., J.M.P. and N.M.). This project also received funding from Fundació la Marató de TV3 201910-31 (N.M.) and 202125-30 (N.M. and A.M.). J.M.P. received funding from the Austrian Academy of Sciences, the Medical University of Vienna, the Vienna Science and Technology Fund WWTF (10.47379/EICOV20002), the Fundacio La Marato de TV3 (202125-31), the T. von Zastrow foundation and the Canada 150 Research Chairs Program F18-01336. We also acknowledge funding from the German Federal Ministry of Education and Research (BMBF) under the project 'Microbial Stargazing - Erforschung von Resilienzmechanismen von Mikroben und Menschen' (Ref. 01KX2324) (J.M.P.). This study was also funded by Instituto de Salud Carlos III (ISCIII) through the Biobanks and Biomodels Platform and co-funded by the European Union (PTC20/00013 and 10 PTC20/00130) (N.M.), received funding from the University of Padova PRID-2017 (C.S.) and was supported by EU funding within the MUR PNRR Extended Partnership initiative on Emerging Infectious Diseases (Project no. PE00000007, INF-ACT) (C.S.). S.C.W. was supported by the Swedish Society for Medical Research (PD20-0153). V.M.L. was supported by the Swedish Research Council (grant agreement numbers 2019-01837 and 2021-02801), the EU/EFPIA/OICR/McGill/KTH/Diamond Innovative Medicines Initiative 2 Joint Undertaking (EUbOPEN grant number 875510), and the Robert Bosch Foundation, Stuttgart, Germany. F.W. was supported by the LOEWE Center DRUID of the Land Hessen. This research was also supported in part by the Division of Intramural Research, NIAID/NIH (D.W.H., H.F.). The funders had no role in study design, data collection and analysis, decision to publish or preparation of the manuscript.

## Author contributions

V.M.M. performed all the experiments involving viruses, designed most of the experiments, cultivated the clinical CCHFV isolate as well as all CCHFV used in the study and wrote the manuscript. M.D. developed and produced the cell library. S.C.W. designed and set up the BRET experiment with the help of I.P. and S.Y. M.J.K. prepared the knockout cells (Vero, A549 and NC8) and S.W.P. produced the blood vessel organoids. S. Appelberg performed the mouse BSL-4 experiments with the help of V.M.M., S.D., S. Abdurahman, D.W.H. and S.B. A.I. and T.A. ran the affinity data and did the corresponding analysis. H.K. performed the RT–qPCR experiments. M.M. and C.S. developed and produced VSV-CCHF_G. S.D. runned ApoE antibody neutralization assays. G.M. performed sequencing analysis of resistant clones. E.E. developed the standard RNAs for CCHFV. B.B. and N.E. sampled the serum of the CCHF patients. V.M.L., C.S., N.M., F.W., H.F., M.H., J.M.P. and A.M. helped with manuscript editing, and J.M.P and A.M. with the design of experiments.

## Funding

## Competing interests

J.M.P. is a founder and shareholder of JLP. V.M.L. is co-founder, CEO and shareholder of HepaPredict AB, as well as co-founder and shareholder of PersoMedix AB, and discloses consultancy work for Enginzyme AB. M.H. is co-founder, CEO and shareholder of Acus Laboratories GmbH and CSO of JLP Health GmbH. M.D. is an employee of Acus Laboratories GmbH and JLP Health GmbH. A.I. is an employee of Attana AB and T.A. is CSO of Attana AB. A patent application has been filed (PCT application, 2023, European patent EP 23 174 811.2). All other authors declare no competing interests.

## Additional information

[1]Unit of Clinical Microbiology, Department of Laboratory Medicine, Karolinska Institute and Karolinska University Hospital, Stockholm, Sweden. [2]Public Health Agency of Sweden, Solna, Sweden. [3]Department of Physiology and Pharmacology, Karolinska Institutet, Stockholm, Sweden. [4]Acus Laboratories GmbH, Cologne, Germany. [5]JLP Health GmbH, Vienna, Austria. [6]IMBA, Institute of Molecular Biotechnology of the Austrian Academy of Science, Vienna, Austria. [7]Vienna Biocenter PhD Program, a Doctoral School of the University of Vienna and the Medical University of Vienna, Vienna, Austria. [8]Attana AB, Stockholm, Sweden. [9]National Veterinary Institute, Uppsala, Sweden. [10]Department of Molecular Medicine, University of Padova, Padova, Italy. [11]Department of Nutrition and Dietetics, Faculty of Health Sciences, Sivas Cumhuriyet University, Sivas, Turkey. [12]University Tübingen, Tübingen, Germany. [13]Dr. Margarete Fischer-Bosch Institute of Clinical Pharmacology, Stuttgart, Germany. [14]Department of Infectious Diseases and Clinical Microbiology, Medical Faculty, Cumhuriyet University, Sivas, Turkey. [15]Institute for Virology, FB10-Veterinary Medicine, Justus-Liebig University, Gießen, Germany. [16]University of Barcelona, Barcelona, Spain. [17]Pluripotency for Organ Regeneration, Institute for Bioengineering of Catalonia (IBEC), The Barcelona Institute of Science and Technology (BIST), Barcelona, Spain. [18]Catalan Institution for Research and Advanced Studies (ICREA), Barcelona, Spain. [19]Rocky Mountain Laboratories, NIAID/NIH, Hamilton, MT, USA. [20]Department of Laboratory Medicine, Medical University of Vienna, Vienna, Austria. [21]Helmholtz Centre for Infection Research, Braunschweig, Germany. [22]Department of Medical Genetics, Life Sciences Institute, University of British Columbia, Vancouver, British Columbia, Canada. [23]These authors contributed equally: Shane C. Wright, Matheus Dyczynski. ✉e-mail: josef.penninger@helmholtz-hzi.de; ali.mirazimi@ki.se

**Extended Data Table 1 | Sequencing data**

| SampleID | chrom | pos | Mut | sums | freq.(%) | genes | transcript length | ANN |
|---|---|---|---|---|---|---|---|---|
| CCRH_5 | chr9 | 21538165 | C>T | 175 | 100 | *Ldlr* | 4627 | T-stop_gained |
| CCRH_8 | chr9 | 21536769 | T>A | 188 | 100 | *Ldlr* | 4627 | A-stop_gained |
| CCRH_10 | chr9 | 2544230 | T>A | 144 | 100 | *Ldlr* | 4627 | A-missense_variant |

Deep exon sequencing identified three distinct single point mutations in the *Ldlr* gene in three resistant clones (5, 8, 10) resulting in stop codons or a missense mutation.

**Extended Data Table 2 | Affinity data**

| Averaged Kinetic Parameters | | | | |
|---|---|---|---|---|
| | $k_{a1}$ ($10^4 M^{-1} s^{-1}$) | $k_{d1}$ ($10^{-3} s^{-1}$) | $k_{D1}$ (e-9) | $B_{max1}$ (Hz) |
| LDL | 25.5 | 1.12 | 3 | 70 |
| CCHFV Gc | 1.3 | 0.059 | 3.3 | 64 |
| Gc + Gn | 2.1 | 0.00867 | 0.272 (272pM) | 7 |

The QCM data were assessed and analysed using the Evaluation (Attana AB) and TraceDrawer software (Ridgeview Instruments), employing 1:1 or 1:2 binding models to calculate kinetic parameters, including rate constants (ka, kd), dissociation equilibrium constant (KD), and maximum binding capacity (Bmax).

**Extended Data Table 3 | Scoring system for mice study**

| Parameter | Points | Observations |
|---|---|---|
| General condition | 0.0 | Alert, Live, active, reacts to their surroundings |
| | 0.1 | Slow, weak, less active |
| | 0.4 | Immobile, limited or no voluntary movement, lying motionless |
| The eyes | 0.0 | Clear and clean eyes |
| | 0.1 | Slight discharge around eyes and nose |
| | 0.4 | Discharge on the face and/or legs and paws. Swollen eyes |
| Movement and posture | 0.0 | Normal |
| | 0.1 | Cannot fully coordinate movements |
| | 0.4 | One or more of the following: Marked incoordination, hunched posture or back, lying motionless, severe lameness |
| Piloerection | 0.0 | Fur smooth and well-groomed |
| | 0.1 | Mild piloerection |
| | 0.4 | Severe piloerection |

Mice infected with CCHFV reaching the human Endpoint score fixed to 0.8 points according to Karolinska Institutet were euthanized.

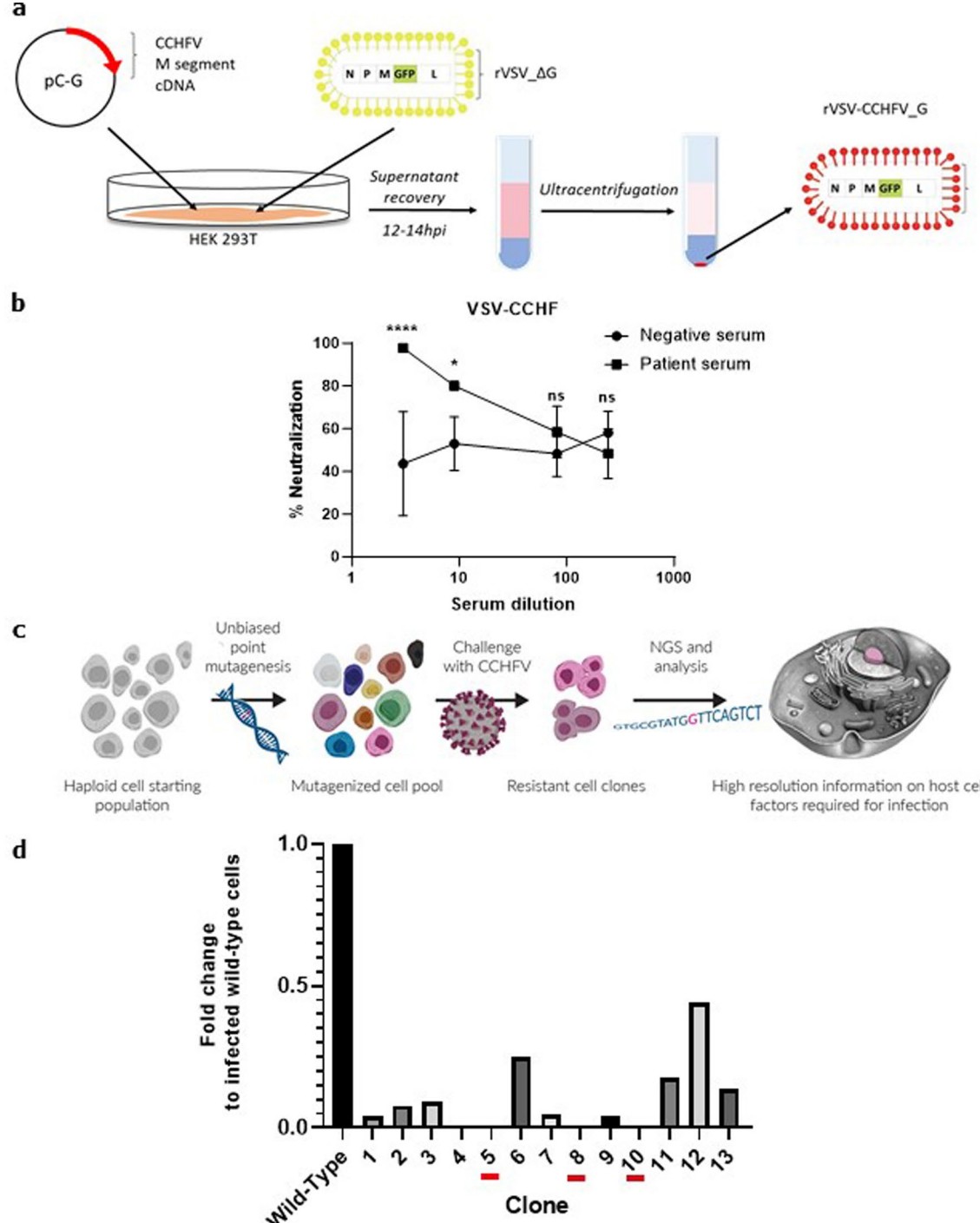

**Extended Data Fig. 1 | Generation of VSV-CCHF_G and haploid cells screening.** **a**, Schematic representation of the methods used to produce the VSV-CCHF_G pseudotype virus in HEK293T cells. **b**, To validate the functionality of the glycoprotein complex in VSV-CCHF, a sero-neutralization was conducted using a serum from vaccinated person or a control serum. Data represent mean ± SD. n = 4. Two-way ANOVA. *p < 0.05, ****p < 0.001, ns: non significant. n = 3 independent experiments. Exact p-values are available in Source data.

**c**, Scheme of the haploid cell screening system. NGS, Next Generation Sequencing **d**, Validation of resistant clones obtained in the primary haploid screen with VSV-CCHF_G. Each clone (1-13) was isolated, amplified and assessed for infection with the CCHFV IbAR10200 laboratory strain (MOI 0.1). The data show the level of infection for each clone compared to wild-type haploid cells (AN3-12) as determined by RT-PCR for CCHFV and RNase P RNA 24hpi.

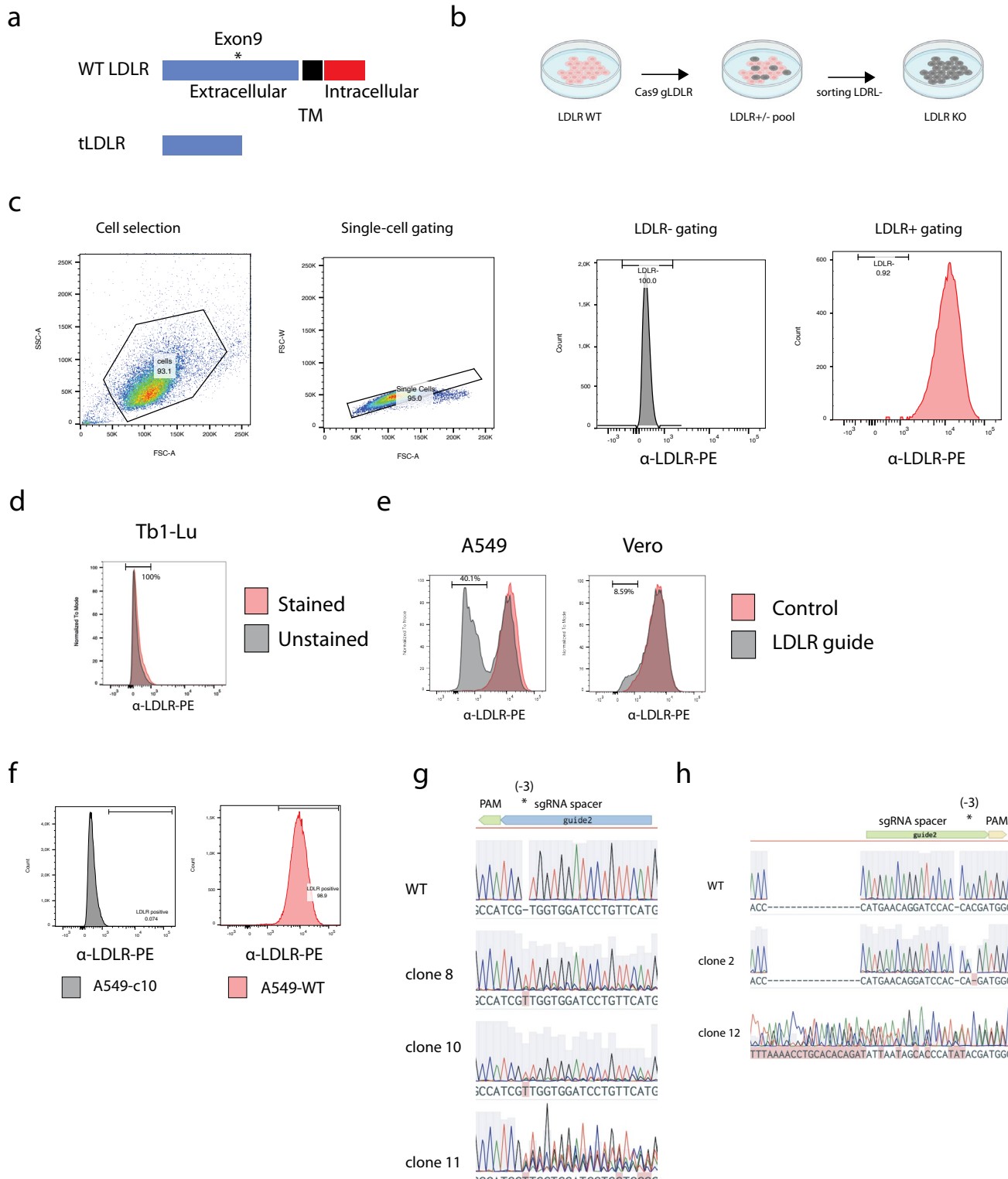

**Extended Data Fig. 2 | See next page for caption.**

**Extended Data Fig. 2 | Generation and validation of knockouts in A549 and Vero cells. a**, Schematic of CRISPR-Cas9 editing strategy. The extracellular region of LDLR was targeted, leading to putative N-terminally truncated proteins not displayed on the cell surface for entry. **b**, Schematic of editing and α-LDLR sorting procedure. **c**, Gating strategy and PE intensity from α-LDLR-PE staining are shown. α-LDLR-PE staining was evaluated on single cells. Event densities were smoothened and are displayed as absolute counts or as counts normalization to the mode. Numbers indicate the percentage of single cells defined as α-LDLR-PE negative. **d**, Non-reactive and stained Tb1-Lu cells were used as negative control. PE intensity from α-LDLR-PE staining is shown. Event density was smoothened by normalization to the mode. **e**, Bulk sorting after transfection and transient Puromycin selection of α-LDLR-PE stained A549 or Vero cells, edited or unmodified (control) via CRISPR-Cas9. Event densities were smoothened and are displayed as counts normalization to the mode. **f**, Flow-cytometry result from A549-wild type and edited A549 clone 10 cells. PE intensity from α-LDLR-PE staining is shown. Event density was smoothened by normalization to the mode. Numbers indicate the percentage of single cells defined as α-LDLR-PE positive for A549 clone 10 and unmodified WT cells. Event densities were smoothened and are displayed as absolute counts. **g**, Sanger sequencing of PCR products from the LDLR genomic locus CRISPR-Cas9 editing site for A549 clones 8, 10 and 11 alongside unmodified wild-type cells are shown (via benchling.com alignment). The sgRNA spacer, PAM and expected Cas9 editing site (3 base-pairs downstream of PAM sequence) are shown above the sequencing traces. **h**, Same as shown for f, but for Vero cell clones C2, C12 and wild-type cells.

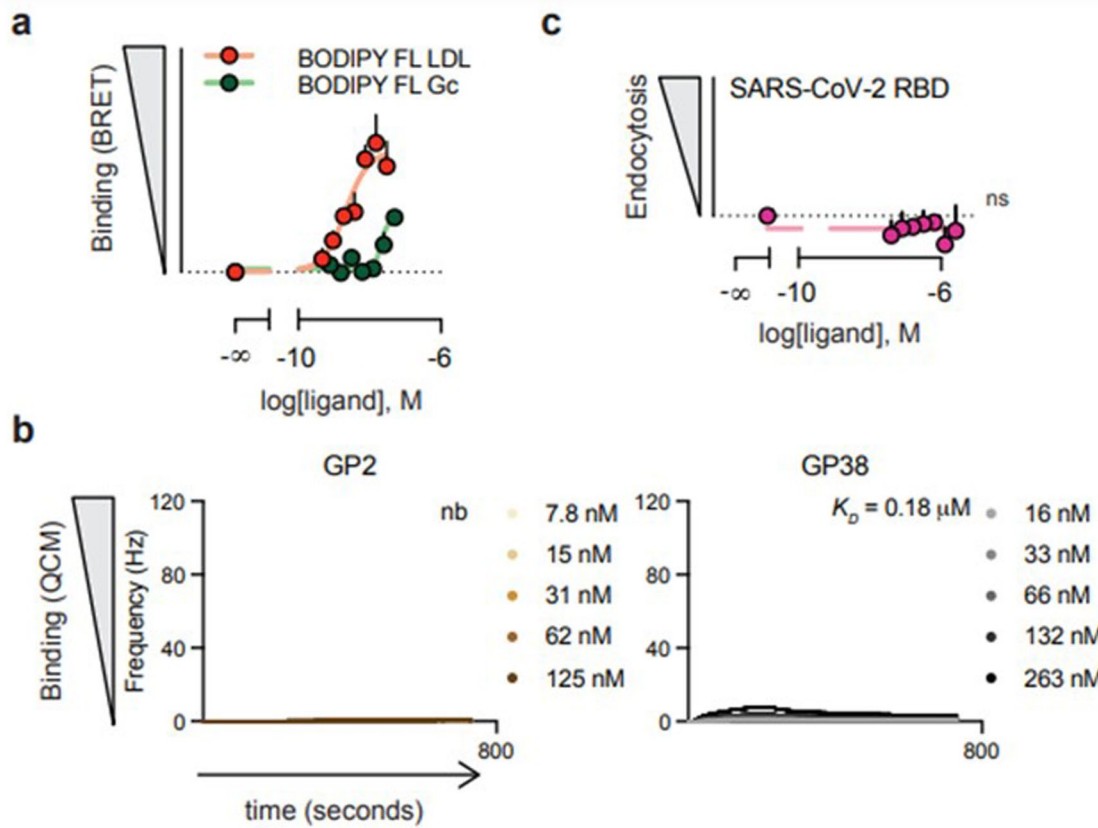

**Extended Data Fig. 3 | Additional measures of ligand selectivity at LDLR and controls for BRET and QCM experiments. a**, Cells expressing Nluc-LDLR (donor) were stimulated with vehicle or increasing concentrations of BODIPY-FL-labelled LDL or Gc (acceptor) for 90 min during which the BRET was measured continuously. Data are represented as the mean area under the curve ± SEM (n = 5 biologically independent samples). **b**, Kinetic QCM experiments monitoring the interaction between G2 (Toscana virus) or GP38 with the extracellular domain of LDLR. Data are presented as mean ± s.e.m. of n = 3 independent experiments. **c**, SARS-CoV-2 RBD does not induce internalization of LDLR. Data are represented as the mean ± SEM (n = 3). Binding and internalization were assessed by comparing the top and bottom parameters from non-linear regression in the extra sum-of-squares F-test (P < 0.05). ns non-significant (one-tailed extra sum-of-squares F test).

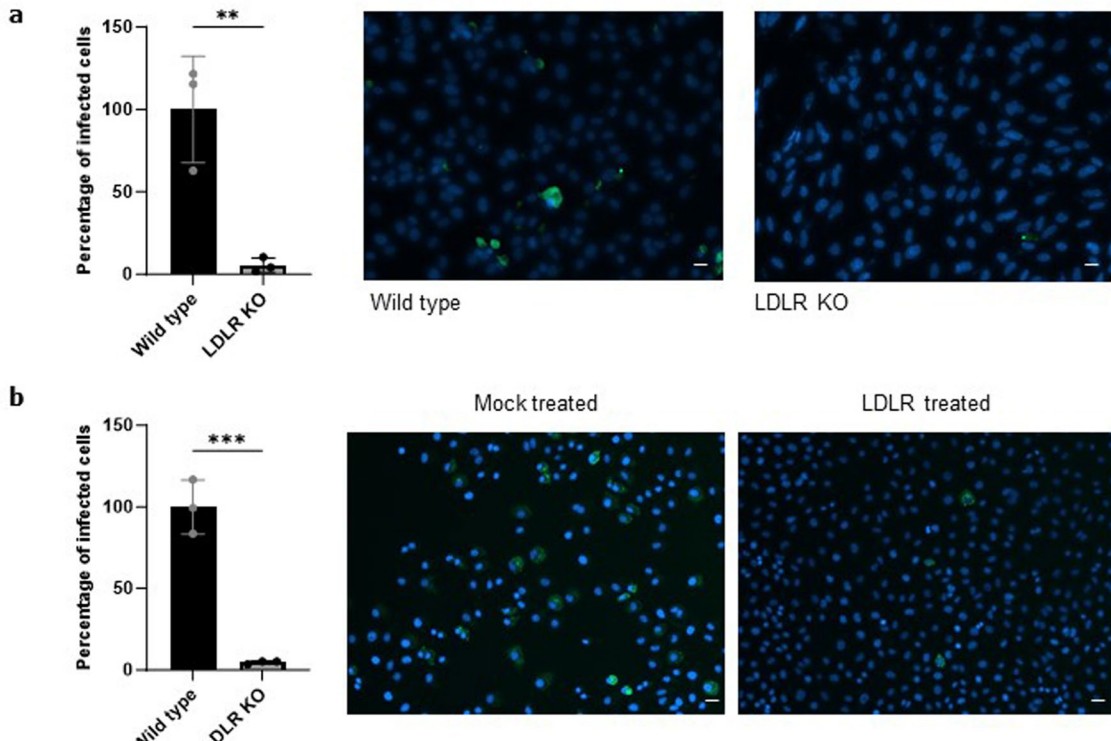

**Extended Data Fig. 4 | Immunoflurorescence assays. a**, Immunofluorescence staining of CCHFV in wild-type and *LDLR* KO cells. *P* values were calculating using two tailed student *t*-test. Data are presented as mean values +/- SD. **P < 0.01. n = 3 independent experiments. Scale bar: 10 μm **b**, Immunofluorescence staining of CCHFV in SW13 cells infected with CCHFV mock or sLDLR treated. *P* values were calculating using two tailed student *t*-test. ***P < 0.001. Data are presented as mean values +/- SD. n = 3 independent experiments. Scale bar: 10 μm. All Pictures are representative of 3 wells from independent experiments. Exact p-values are available in Source data.

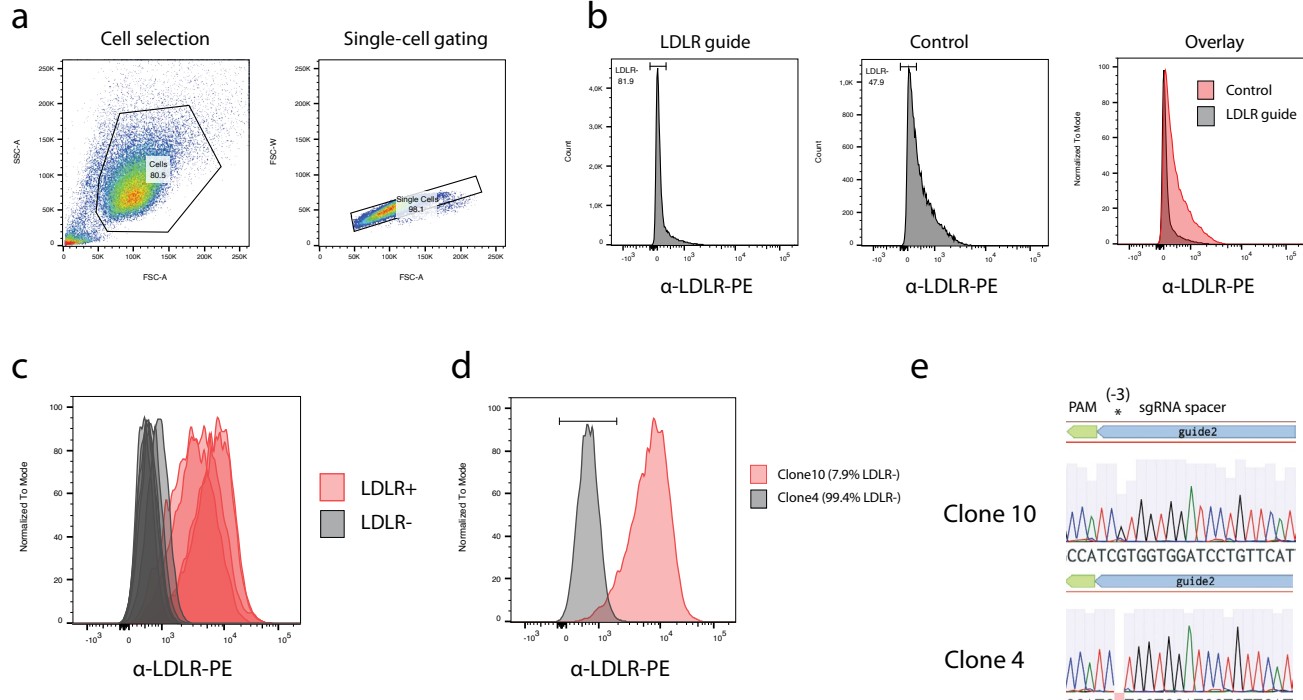

**Extended Data Fig. 5 | Creation and validation of NC8 cells knocked out for LDLR. a**, Gating strategy and PE intensity from α-LDLR-PE staining are shown. α-LDLR-PE staining was evaluated on single cells. **b**, Sorting results from bulk NC8 iPSC after Cas9 LDLR editing. PE intensity from α-LDLR-PE staining is shown for cells targeted with an LDLR guide RNA or for unmodified control cells. Event densities were smoothened and are displayed as absolute counts or as counts normalization to the mode. **c**, Qualitative flow-cytometry result of selected clones stained with an α-LDLR-PE antibody. Shown is the PE intensity from α-LDLR-PE staining from gated single cells of LDLR- or LDLR+ iPSC clones.

**d**, Flow-cytometry result of the studied LDLR-KO or wild-type LDLR iPSC clones (clone 4, clone 10). Shown is the overlayed mode-normalized density of PE intensity from α-LDLR-PE staining for clone 10 and 4. The legend percentages indicate the fraction of α-LDLR-PE negative stained single cells. **e**, Sanger sequencing of PCR product from the LDLR genomic locus CRISPR-Cas9 editing site for NC8 iPSC clones 10 and clone 4 are shown. The sgRNA spacer, PAM and expected Cas9 editing site (3 base-pairs downstream of PAM sequence) are shown above the sequencing traces.

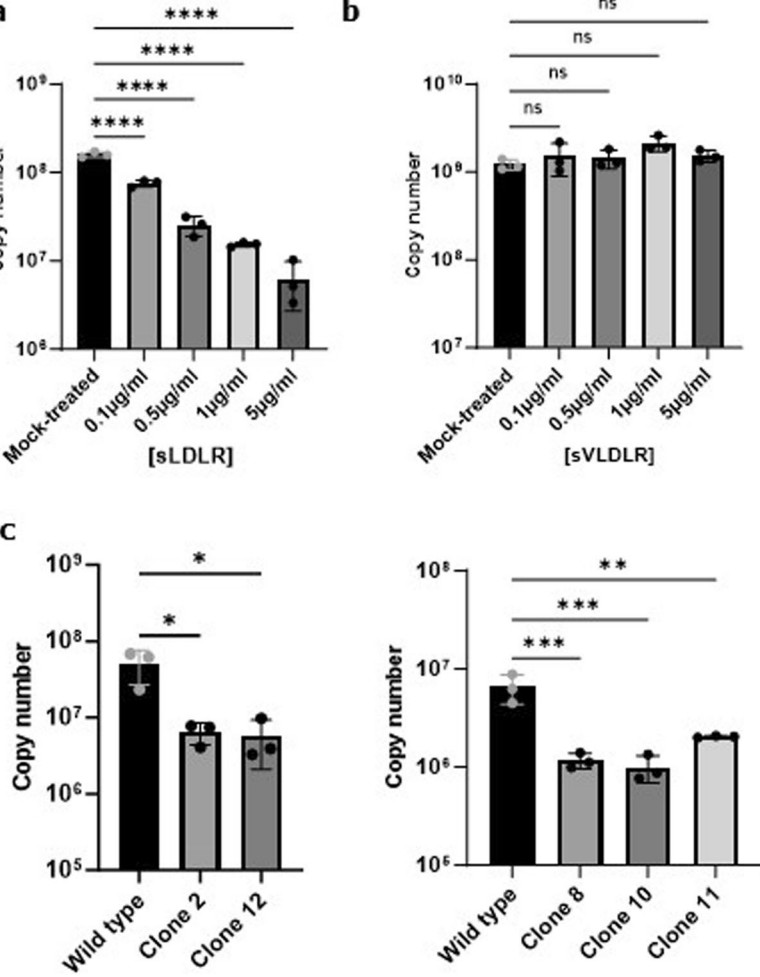

**Extended Data Fig. 6 | Validation of LDLR with a CCHFV patient isolate.** CCHFV was isolated from the serum of a Turkish patient and this clinical isolate used for all subsequent experiment in Fig. 6. **a**, **b**, Levels of CCHFV infections of SW13 cells treated (MOI 0.01, 24hpi) with the indicated concentrations of sLDLR, and sVLDLR. **a**, sLDLR. **b**, sVLDLR **c**, Levels of infection with clinical CCHFV in wild type and *LDLR* KO (clones C2 and C12) Vero cells and in wild type and *LDLR* KO (clones C8, C10 and C11) A549 cells (MOI 0.1, 24hpi). Graphs show mean value ± SD. n = 3 independent experiments. *P* values were calculating using One-way ANOVA. *P < 0.05, **P < 0.01; *** P < 0.001; **** P < 0.0001. Non significant: p > 0.05. Exact p-values are available in Source data.

Joseph Penninger

# Reporting Summary

## Statistics

For all statistical analyses, confirm that the following items are present in the figure legend, table legend, main text, or Methods section.

| n/a | Confirmed | |
|---|---|---|
| ☐ | ☒ | The exact sample size (*n*) for each experimental group/condition, given as a discrete number and unit of measurement |
| ☐ | ☒ | A statement on whether measurements were taken from distinct samples or whether the same sample was measured repeatedly |
| ☐ | ☒ | The statistical test(s) used AND whether they are one- or two-sided *Only common tests should be described solely by name; describe more complex techniques in the Methods section.* |
| ☒ | ☐ | A description of all covariates tested |
| ☐ | ☒ | A description of any assumptions or corrections, such as tests of normality and adjustment for multiple comparisons |
| ☐ | ☒ | A full description of the statistical parameters including central tendency (e.g. means) or other basic estimates (e.g. regression coefficient) AND variation (e.g. standard deviation) or associated estimates of uncertainty (e.g. confidence intervals) |
| ☐ | ☒ | For null hypothesis testing, the test statistic (e.g. *F*, *t*, *r*) with confidence intervals, effect sizes, degrees of freedom and *P* value noted *Give P values as exact values whenever suitable.* |
| ☒ | ☐ | For Bayesian analysis, information on the choice of priors and Markov chain Monte Carlo settings |
| ☒ | ☐ | For hierarchical and complex designs, identification of the appropriate level for tests and full reporting of outcomes |
| ☒ | ☐ | Estimates of effect sizes (e.g. Cohen's *d*, Pearson's *r*), indicating how they were calculated |

*Our web collection on statistics for biologists contains articles on many of the points above.*

## Software and code

Policy information about availability of computer code

| | |
|---|---|
| Data collection | Attester Software |
| Data analysis | GraphPad Prism (version 9.4.1), GATK (version 4.5.0.0), snpEff (version 5.2), Attester Evaluation software, TraceDrawer software (version1.9.1), BD FACSDiva (version 9.0.1) , FlowJo (version 10.8.1) |

For manuscripts utilizing custom algorithms or software that are central to the research but not yet described in published literature, software must be made available to editors and reviewers. We strongly encourage code deposition in a community repository (e.g. GitHub). See the Nature Portfolio guidelines for submitting code & software for further information.

## Data

Policy information about availability of data

All manuscripts must include a data availability statement. This statement should provide the following information, where applicable:

- Accession codes, unique identifiers, or web links for publicly available datasets
- A description of any restrictions on data availability
- For clinical datasets or third party data, please ensure that the statement adheres to our policy

Sequencing raw data in form of .vcf files is accessible on https://github.com/GMichlits/CCHFV-LDLR. The data generated in this study are provided in the Supplementary Information/Source Data file. Source data are provided with this paper.

## Human research participants

Policy information about studies involving human research participants and Sex and Gender in Research.

| Reporting on sex and gender | N/A |
| Population characteristics | N/A |
| Recruitment | N/A |
| Ethics oversight | N/A |

Note that full information on the approval of the study protocol must also be provided in the manuscript.

# Field-specific reporting

Please select the one below that is the best fit for your research. If you are not sure, read the appropriate sections before making your selection.

☒ Life sciences ☐ Behavioural & social sciences ☐ Ecological, evolutionary & environmental sciences

For a reference copy of the document with all sections, see nature.com/documents/nr-reporting-summary-flat.pdf

# Life sciences study design

All studies must disclose on these points even when the disclosure is negative.

| Sample size | For all experiments, 3 biological independent experiment for statistical relevance (One-Way ANOVA, student t-test, Kruskal-Wallis test with uncorrected Dunn's test) <br> For mice experiments, 12 mice per group, more than the number of mice usually used for these type of experiment (8 mice) |
| Data exclusions | No data excluded |
| Replication | 3 independent experiments, consistent data between the experiments, all replication attempt were successful |
| Randomization | For in vitro study, randomisation not possible as either the same cells are treated differently, or the cells are genetically different: Thus, a random distribution of the samples was not possible. <br> For in vivo experiments, randomisation was not possible as mice in the 2 groups were genetically differents |
| Blinding | For all experiments (in vitro and in vivo), the samples curation was not blinded but the analysis were blinded (the experimentors that runned the experiments and the ones that analyzed the data were different, the samples not showing any information that could allow their identification during analysis). |

# Reporting for specific materials, systems and methods

We require information from authors about some types of materials, experimental systems and methods used in many studies. Here, indicate whether each material, system or method listed is relevant to your study. If you are not sure if a list item applies to your research, read the appropriate section before selecting a response.

## Materials & experimental systems

| n/a | Involved in the study |
| ☐ | ☒ Antibodies |
| ☐ | ☒ Eukaryotic cell lines |
| ☒ | ☐ Palaeontology and archaeology |
| ☐ | ☒ Animals and other organisms |
| ☒ | ☐ Clinical data |
| ☐ | ☒ Dual use research of concern |

## Methods

| n/a | Involved in the study |
| ☒ | ☐ ChIP-seq |
| ☐ | ☒ Flow cytometry |
| ☒ | ☐ MRI-based neuroimaging |

## Antibodies

| Antibodies used | Anti-IFN type I receptor antibody (MAR1-5A3) (MAR1-5A3 [5A3]; Leinco Technologies, Inc.). VSV-M [23H12] antibody, EB0011, |

| Antibodies used | Kerafast. Goat anti-Mouse IgG (H+L) Cross-Adsorbed Secondary Antibody, Alexa Fluor™ 488, A11001, ThermoFisher. α-LDLR-PE Antibody, R&D Systems FAB2148P. ApoE antibody (Sigma,  #AB947) |
| --- | --- |
| Validation | Validation of VSV-M antibody was done for western-blot by Kerafast.  α-LDLR-PE antibody was validated for Flow cytometry against human LDLR by R&D system. The ApoE antibody was validated for neutralization in a previous study (Tréguier Y. et al, Virol J., 2022). The anti-interferon antibody was used in a previous study (Garrison A. et al, Plos Neg. Trop. Dis., 2017) |

# Eukaryotic cell lines

Policy information about cell lines and Sex and Gender in Research

| Cell line source(s) | HEK293 (ATCC®, CRL-1573), HEK293T/17 (HEK293T, ATCC® CRL-11268™), A549 (ATCC® CCL-185) and Vero cells (ATCC® CCL-81). Haploid mouse Stem-Cells (mSCs, clone AN3-12) from IMBA, HepG2 (Abcam, AB275467), HepG2 ApoE KO (Abcam, AB280875), SW13 (ATCC, CCL-105), Bat Tb-1 Lu cells (ATCC, CCL-88) |
| --- | --- |
| Authentication | AN3-12 were authenticated by Haplobank (IMBA, Vienna). A549, HEK293, HEK293T and Vero cells were authenticated by STR profiling. HepG2 wt and ApoE KO were authentified by Abcam. Bat Tb-1 Lu were authenticated by ATCC. |
| Mycoplasma contamination | Cell lines were tested for mycoplasma contamination with negative results |
| Commonly misidentified lines (See ICLAC register) | No commonly misidified cell lines was used in thi sstudy |

# Animals and other research organisms

Policy information about studies involving animals; ARRIVE guidelines recommended for reporting animal research, and Sex and Gender in Research

| Laboratory animals | 10 weeks old C57BL/6J mice (Charles River, Germany) and B6.129S7Ldlrtm1Her/J (LdLr KO), stock#002207, Jackson Laboratory, USA. |
| --- | --- |
| Wild animals | No wild animals were used in this study |
| Reporting on sex | Female mice. Sex was considered but female were choosen to avoid mice aggressiveness, more difficult to address in BSL4 conditions. |
| Field-collected samples | The study didn't involve fiels-collected samples |
| Ethics oversight | Stockholm Ethical Committee for animal research approved the research. |

Note that full information on the approval of the study protocol must also be provided in the manuscript.

# Dual use research of concern

Policy information about dual use research of concern

## Hazards

Could the accidental, deliberate or reckless misuse of agents or technologies generated in the work, or the application of information presented in the manuscript, pose a threat to:

| No | Yes |
| --- | --- |
| ☐ | ☒ Public health |
| ☒ | ☐ National security |
| ☒ | ☐ Crops and/or livestock |
| ☒ | ☐ Ecosystems |
| ☒ | ☐ Any other significant area |

| Hazards | Crimean-Congo Hemorrhagic Fever Virus |
| --- | --- |

For examples of agents subject to oversight, see the United States Government Policy for Institutional Oversight of Life Sciences Dual Use Research of Concern.

## Experiments of concern

Does the work involve any of these experiments of concern:

| No | Yes | |
|---|---|---|
| ☒ | ☐ | Demonstrate how to render a vaccine ineffective |
| ☒ | ☐ | Confer resistance to therapeutically useful antibiotics or antiviral agents |
| ☒ | ☐ | Enhance the virulence of a pathogen or render a nonpathogen virulent |
| ☒ | ☐ | Increase transmissibility of a pathogen |
| ☒ | ☐ | Alter the host range of a pathogen |
| ☒ | ☐ | Enable evasion of diagnostic/detection modalities |
| ☒ | ☐ | Enable the weaponization of a biological agent or toxin |
| ☒ | ☐ | Any other potentially harmful combination of experiments and agents |

## Precautions and benefits

| | |
|---|---|
| Biosecurity precautions | All our experiments involving VSV-CCH_G were done in biosafety level 2 laboratory and experiments involving CCHFV were done in biosafety level 4 laboratory in compliance with the Swedish Public Health Agency guidelines (Folkhälsomyndigheten, Stockholm). |
| Biosecurity oversight | Infections of cells are regulated the Swedish Public Health Agency SOPs reviewed by internal Biorisk committee. The internal Biorisk committe also reviewed and approved specifically the mice experiment described in this study |
| Benefits | Developement of antivirals to treat CCHFV infected patients. |
| Communication benefits | There is no risk communicating the information given in this manuscript |

# Flow Cytometry

## Plots

Confirm that:

☒ The axis labels state the marker and fluorochrome used (e.g. CD4-FITC).

☒ The axis scales are clearly visible. Include numbers along axes only for bottom left plot of group (a 'group' is an analysis of identical markers).

☒ All plots are contour plots with outliers or pseudocolor plots.

☒ A numerical value for number of cells or percentage (with statistics) is provided.

## Methodology

| | |
|---|---|
| Sample preparation | Cells were dissociated with 500μl with TrypLE Express enzyme solution (Gibco) for 5 minutes and collected in FACS Buffer (D-PBS containing 5% FBS). After one wash with FACS Buffer, 10μl of α-LDLR-PE Antibody (R&D Systems FAB2148P) per 1E+6 cells was added and stained for 1h on ice in the dark. After one hour of staining, cells were collected by centrifugation and washed twice in FACS Buffer. Finally, cells were resuspended in 1ml of FACS Buffer |
| Instrument | Sorting was done on a FACS Aria III sorted. Flow cytometry was performed on a FACS LSR Fortessa instrument. |
| Software | Data was analyzed during sorting with BD FACSDiva (version 9.0.1) and re-analyzed for plotting data presented in this manuscript using FlowJo. |
| Cell population abundance | LDLR-negative, edited A549 cells that were sorted comprised 40.1% of single-cells (34.6% of total after cell, single-cell and LDLR-staining gating). LDLR-negative, edited Vero cells comprised 8.59% of single-cells (7.59% of total after cell, single-cell and LDLR-staining gating). LDLR-negative, edited NC8 iPSC that were sorted comprised 81.9% of single-cells (62.6% of total after cell, single-cell and LDLR-staining gating). The non-edited LDLR-negative single cell fractions of A549, Vero and NC8 iPSC were 0.92% 2.32%, and 47.9% respectively. |
| Gating strategy | Forward and side-scatter were used to define cells excluding debri and larger aggregates. Forward scatter area versus height was then used to define single cells. α-LDLR-PE-staining density was then plotted and LDLR-negative cells selected based on unmodified, stained cells. These cells were then sorted into a 96-well plate. |

☒ Tick this box to confirm that a figure exemplifying the gating strategy is provided in the Supplementary Information.

