## [Peer Review File · Nature Microbiology]

Peer Review Information

Journal: Nature Microbiology

Manuscript Title: Crimean-Congo Hemorrhagic Fever virus uses LDLR to bind and enter host cells

Corresponding author name(s): Professor Ali Mirazimi

Editorial Notes:

This manuscript has been previously reviewed at another journal. This document only contains reviewer comments, rebuttal and decision letters for versions considered at Nature Microbiology. Mentions of prior referee reports have been redacted

Reviewer Comments & Decisions:Decision Letter, initial version:

Message: 20th July 2023

Dear Ali,

Thank you for submitting your paper "LDLR as a critical receptor for Crimean-Congo Hemorrhagic Fever virus" to Nature Microbiology. I'm writing to let you know that we are very interested in the possibility of publishing your study in Nature Microbiology, but would like to consider your response to the raised concerns by the reviewers in the form of a fully-revised manuscript and a point-by-point response to all the reviewers comments, before we make a final decision on publication.

In particular, please address following points raised by the reviewers:

1. Characterize the suitability to use pseudotyped viruses to study CCHFV entry (i.e. Gn and Gc have the correct conformation).
2. Provide information on the suitability to use blood vessel organoids your to study CCHFV
3. Investigate the role of Gn during viral entry via LDLR
4. Use orthogonal approaches to confirm LDLR as entry receptor and to quantify virus entry for CCHFV via LDLR as suggested by the reviewers.
5. Adjust sample numbers, time to analyze post-infection, include additional assays to look at pathogenesis and define disease scoring, in your vivo experiments
6. Confirm expression levels of LDLR in KD and mutant setups.

The remaining issues should be straightforward to address.

I should stress that we would be unlikely to go back to the reviewers in the absence of substantial changes to address the technical concerns made by all reviewers.

Please include a data availability statement as a separate section after Methods but before references, under the heading "Data Availability". This section should inform readers about the availability of the data used to support the conclusions of your study. This information

3includes accession codes to public repositories (data banks for protein, DNA or RNA sequences, microarray, proteomics data etc...), references to source data published alongside the paper, unique identifiers such as URLs to data repository entries, or data set DOIs, and any other statement about data availability. At a minimum, you should include the following statement: "The data that support the findings of this study are available from the corresponding author upon request", mentioning any restrictions on availability. If DOIs are provided, we also strongly encourage including these in the Reference list (authors, title, publisher (repository name), identifier, year). For more guidance on how to write this section please see:
<http://www.nature.com/authors/policies/data/data-availability-statements-data-citations.pdf>

* If you have not done so already we suggest that you begin to revise your manuscript so that it conforms to our Article format instructions at <http://www.nature.com/nmicrobiol/info/final-submission>. Refer also to any guidelines provided in this letter.

When submitting the revised version of your manuscript, please pay close attention to our [href="https://www.nature.com/nature-portfolio/editorial-policies/image-integrity">Digital Image Integrity Guidelines](https://www.nature.com/nature-portfolio/editorial-policies/image-integrity). and to the following points below:

Note: This url links to your confidential homepage and associated information about manuscripts you may have submitted or be reviewing for us. If you wish to forward this e-mail to co-authors, please delete this link to your homepage first.

Nature Microbiology is committed to improving transparency in authorship. As part of our efforts in this direction, we are now requesting that all authors identified as 'corresponding author' on published papers create and link their Open Researcher and Contributor Identifier (ORCID) with their account on the Manuscript Tracking System (MTS), prior to acceptance. This applies to primary research papers only. ORCID helps the scientific community achieve unambiguous attribution of all scholarly contributions. You can create and link your ORCID from the home page of the MTS by clicking on 'Modify my Springer Nature account'. For more information please visit please visit www.springernature.com/orcid.

Please let me know about the timeline you anticipate to submit a fully revised manuscript. If the revisions will take longer than 6 months we will reassess the novelty aspect of the manuscript.

Should further experimental data allow you to address these criticisms, we would be happy to look at a revised manuscript.

Yours sincerely,

Author Rebuttal to Initial comments

Answers to reviewers

Referee #1 (Remarks to the Author):

In this manuscript by Monteil et al., the authors propose that Low Density Lipoprotein Receptor (LDLR) is a critical receptor for Crimean-Congo hemorrhagic fever virus (CCHFV). Using genetic screening in murine haploid cells deploying both pseudotype VSV-CCHFV and wild-type CCHFV they identify LDLR as a receptor for CCHFV. Subsequently, binding was confirmed in African green monkey cells and human lung epithelial cells (reference lab strain and clinical strain of CCHFV). Using biosensor experiments in living cells shows that soluble Gc from CCHFV can bind to LDLR, and soluble LDLR can block CCHFV infection. The findings are corroborated by infecting human blood vessel organoids and LDLR knockout mice.

The data presented here is clearly an important finding. The experimental approach is a tried- and-true approach and yielded fruitful results with live CCHF virus. Nevertheless, the authors need to temper the language as LDLR is one important entry factor out of multiple. Most importantly, although the data presented here indicate that LDLR plays a role in virus entry, the data needs to be more rigorously investigated to define and quantify the role of LDLR as a crucial receptor in CCHFV infection. For instance, using an uncharacterized pseudotype virus or for CCHFV-filed new organoid system to justify some of this approach seems haphazard. A better comparison should have been done with either viral-like particles or viral replicon particles for CCHFV. Better cell lines, specifically those that rely on canonical LDL receptors for function, such as liver cells, would have been much better alternatives for study, both for the receptor and for viral tropism. The in vivo mouse studies

5have major experimental design and interpretation flaws. Thus, additional experiments and controls must be conducted to discern if LDLR is just a cofactor.

General comments on the manuscript:

- The authors focus solely on Gc in their studies and not both glycoproteins Gn and Gc, which form a unit on the virion surface. Curiously, the authors did not look at Gn first, as presumably it is the attachment protein and Gc is the fusion protein. Was Gn also tested, and if not, why? Recent studies in the field have revealed new findings on the surface of the virion. For example, GP38 is also supposedly incorporated in the virion, and Gn/Gc can be former dimers or trimers. By exclusively focusing on the Gc, I am concerned that vital tertiary structures on the virion and its glycoproteins relevant for receptor binding are overlooked (see also below, Gc binding to the receptor).

Response: We agree. We included data on Gn, Gn+Gc as well as GP38 in revised MS (Figure 3 and extended data figure 3 – MS lines 172-212).

- Throughout the manuscript, the authors use “level of infection” as a readout of virus entry. First of all, I have an issue with how this is measured. “Level of Infection” as a readout unit (relative quantification using $\Delta\Delta$ analysis) when it is measuring viral RNA, which is basically genome copies. Few CCHFV publications have used the $\Delta\Delta$ analysis, which is rather unusual. Why was not an absolute RNA quantification in the form of genome copies used just like it was used in animal studies?

Response: We appreciate reviewer feedback. In the revised manuscript, we have now presented our data in terms of CCHFV copy number, providing a clearer implication (fig 2b-c; fig 3e-f; fig 4b and e; fig 5b-c; fig 6; fig 7b,c,f and g; fig 8)

Genome copies can be correlated to infectious units such as PFU if necessary. Secondly, the authors ought to use a second technique to confirm the level of entry. This could be IFA when working with a live virus or GFP expression when using a GFP-labeled VSV pseudotype virus. I don't understand why this was not attempted in the first place.

Response: In revised MS, The data from IFA are now added as Extended data figure 4.

- For all knockdown studies, the authors should show the data for the LDLR knockdown in the cells (at least be a supplemental material).

Response: In our MS, There is no knockdown data. However, the level of LDLR expression in our Knockout cells are already presented in Extended data figure 2 (KO diploid cells) and 5 (KO blood vessels).

Introduction

65: I thought Dengue was the most widespread HFV. **Response:**

This has been addressed in revised MS line 94 68: expanding -> expanding

Response: This has been addressed in revised MS line 98

68: The global warming stance is not justified by publications; please cite for this.

Response: This has been addressed in revised MS line 98

Pseudovirus studies:

- 317: This pseudotype has not been adequately characterized to show that conformationally dependent Gc is incorporated on the surface of the virion. Additionally, characterization is needed for this model. The authors state in line 323 that the VSV-pseudotype was previously published (ref. 56). But this is the wrong reference it does not describe the VSV-Gc pseudotype. **Response: In revised MS, A neutralization data has been added as Extended data figure**

1. And the reference was corrected line 480

- VSV should have been used as a positive control for all in vitro experiments once the authors determined it was LDLR. LDL receptor and its family members are cellular receptors for the vesicular stomatitis virus. Danit Finkelshtein 1, Ariel Werman, Daniela Novick, Sara Barak, Menachem

7Rubinstein DOI: 10.1073/pnas.1214441110. Data shows that the cell can take up naked VSV particles (VSV delta G). If LDLR were critical, there should have been little to no viral RNA in the in vitro studies, in my opinion.

Response: Yes, we agree with the reviewer, why was used VSV as positive control for sLDLR experiment and data was shown in figure 4, showing that LDLR is able to block VSV and CCHFV infection. However, we have add VSV in figure 2a. As reviewer noted in, VSV can also use other LDLR family members to enter the cells. Using it in KO human cells would lead to little to no effect on VSV infection.

- Line 104, why was clone 4 not chosen, although it showed similar results as the selected clones (Fig 1c)? Is it, not an LDLR mutation clone?Response: All clones were sequenced. Clone 4 didn't show mutation in LDLR. Only clones 5, 8 and 10. This is now explained in the text (line 143).

- 88: The novelty of using haploid genetic screens to identify receptors is not groundbreaking and has been done for over the last decade for virus receptor identification.

Response: We have de-emphasized this section by removing the term "unique" line 390

- 104: resistant -> resistance

Response: Now modified line 140

Gc binds to LDLR

- As mentioned above, Gn and Gc form a specific tertiary structure on the virion. If soluble Gc is used, how is it guaranteed that the LDLR sees the structures relevant to receptor binding as seen naturally on the virion surface?

Response: The BRET data indicate that Gc has the capability to bind to LDLR. In our latest set of experiments presented in the revised manuscript (figure 3), Gn-Gc exhibit an even stronger ability to attach to LDLR compared to Gc alone. Additionally, affinity data presented in Figure 3b and c (in revised MS) demonstrate micromolar range affinity for Gc and picomolar range affinity for Gn-Gc. Furthermore, we performed an LDL/CCHFV competition assay (using a CCHFV produced on HepG2 ApoE KO cells to avoid any interference with ApoE binding to LDLR), illustrating that LDL competes with CCHFV for binding to LDLR (Figure 3e).

- 145: This must be shown with other bunyavirus Gc constructs to rule out sticky ligands already implicated for Gc binding (nucleolin and DC-SIGN).

Response: In revised MS, BRET analysis was also done with Toscana G2 protein and doesn't show any attachment (Extended data figure 3).

- 169: Is LRP1 previously defined, if not, it needs to be defined here.

Response: Now addressed in the text (Line 156 and 254)

- Why was this not tried in VLP or VRP's which are more suitable models for infection than uncharacterized pseudotyped viruses?

Response: Our screening system relies on cell death which make VLP and VRP unsuitable. The complete lytic nature of VSV-pseudoviruses for our cells makes them an optimal screening system. In revised MS, we demonstrate the neutralization assay, affirming the favorable conformation of Gn/Gc in VSV-CCHF (Extended Figure 1).

- 178: “as well control” -> “as well as control”

Response: This sentence has been changed line 262 to 268

Infection of blood-vessels

I appreciate the author’s effort to confirm the results of cell cultures in primary human cells/organoids. However, I find using an organoid system not characterized for CCHFV is ill-

advised. The experiment here it's not thorough enough to characterize the system as a tool for the infectivity of CCHFV.

- 189: you need citations to support this claim. It is unclear if the tropism is for the virus for CCHFV, or if the pathologies induced by hemorrhagic fevers, in general, elicit the roles of blood vessels. The author states that the blood vessels are key target cells for viral tropism involved in hemorrhaging, and I agree with them; however, for many hemorrhagic fever viruses, it is still unclear if the hemorrhage is due to viral injury or injury due to pro-inflammatory cytokines or both (Hawman and Feldmann, 2018, doi: 10.12688/f1000research.16189). Furthermore, very little data is available for CCHFV infections of endothelial cells, especially in humans (e.g., what percent become infected). Conversely, a vast body of literature characterizes other target cells, such as hepatocytes, macrophages, dendritic cells, and monocytes. A quick review in the NCBI database (Gene ID: 3949) also revealed to me that hepatocytes, besides lung cells, have the highest level of LDLR expression of any cell type in the body why were those cells not chosen since there is much more information available?

Response: Hepatocytes could be an excellent system. However, complete silence of the gene in these cells are VERY complicated if NOT close to impossible. We would like to see the specificity of LDLR in more complex system, that's why we used human blood vessels organoids as endothelial cells are a target for CCHFV (Burt F.J. et al, Arch. Pathol., 1997; Haddock E et al, Nat. Microbiol., 2018; Andersson I, J.Med.Virol, 2006). It is now better highlighted line 280. The fact that there is less virus in *Ldlr* *-/-* mice also highlight the importance of LDLR in CCHFV infection in liver (figure 5c).

The authors state that “the blood vessel organoids are sometimes detrimental to infection,” and thus, the cells were cultured as monolayers. Does this mean there was infectivity variation in the cell culture? Furthermore, isn't a 3D culture of cells more representative than a monolayer? In Figure 5b, the authors use “Level of Infection” as a readout unit (relative quantification using Delta Delta analysis) when measuring viral RNA, which is basically genome copies. Why was not an absolute RNA quantification in the form of genome copies used just like it was used in the animal studies in the same figure panel?

Response: The challenge with infecting blood-vessel organoids lies in their growth within a collagen/matrigel matrix, rendering the cells less accessible to the virus. To address this, organoids were initially seeded in a 2D format to maintain cellular diversity and enhance virus accessibility. Further clarification on this matter is provided in the text (line 280). Additionally, in response to suggestions, we have modified the data presentation to viral copy numbers (fig 2b-c; fig 3e-f; fig 4b and e; fig 5b-c; fig 6; fig 7b,c,f and g; fig 8).

The data show an RNA difference between WT and KO cells; nevertheless, the KO cells still had a significant amount of RNA. I believe the crux of the whole paper is reflected in the author's statement in line 200: “...though another entry route appears to be operational”. If the virion has multiple modes of getting into the cell, and one of those modes is not accessible/knockout, it is conceivable that the entry is not as efficient. Thus, looking at days 1 and 3 post-infection and extending the timespan would be beneficial.

Response: While acknowledging that LDLR may not be the sole receptor and there are likely multiple receptors for CCHFV, our data unequivocally reveal a substantial 90% decrease in CCHFV levels, underscoring the significance of LDLR as a crucial factor. Wehave reinforced and emphasized this conclusion, particularly highlighting the role of LDLR in virus produced on tick-cells and also in patient via Gn-Gc attachment but also ApoE. (Figure 8, results line 345 to 373 and discussion)

Lastly, as mentioned above, confirming a finding generated with one technique with a second technique is imperative. Thus, an IFA should be performed to also look at what percent of cells (=infectivity rate) are infected express protein.

Response: The IFA data are added as Extended data figure 4, in revised MS

Mouse experiments

The in vivo experiments have some major experimental design flaws.

- First of all, no model characterization of the LDLR knockout mice was done. For example, they are not on the same genetic background (B6.129) as the C57Bl.

Response: We agree, however, C57BL/6 mice are the control suggested by Jackson Laboratory and LDLR KO mouse are extensively characterized and used (Ishibashi S et al, J.Clin.Invest.,1993; Ishibashi S et al, PNAS, 1994; Ishibashi S et al, PNAS, 1994; J.Clin.Invest.,1993; Hertz J et al, PNAS, 1995; Truong TQ et al, Biochim Biophys Acta, 2000; Keren P et al, Diabetes, 2000; Huszar D et al, Arterioscler Thromb Vasc Biol, 2000 etc)

- The dosing schedule of the mAB 5A3 to abolish the IFN response does not follow the published schedule for CCHFV. Thus, the authors need to carry out an endpoint animal study to prove their timing schedule works.

Response: Thx for the comments. The protocol used in this study has been documented in a publication (Hawman D.W. et al, eBioMedicine, 2022). It is now referenced line 685.

In the subsequent mouse study we conducted, we maintained a group of knockout (KO) mice for a survival analysis. The corresponding data are illustrated in Figure 5d.

- Most importantly, the authors cannot make the statement that the mice were protected against the disease if the animals were euthanized on day four, as it is done in line 81. Experiments must be conducted to extend the time to evaluate potential sickness and survival for at least 21 days. Studies have shown that the onset of disease in mice infected with CCHF can be delayed. There was just as much viral genome in some of the LDLR^{-/-} mouse samples as the wild type, which makes me suspect they would have caught up if given a few more days.

Response:

We appreciate the reviewer's comments and would like to express our gratitude. In the second mouse study, we conducted additional experiments, including the retention of a group of knockout (KO) mice for a survival study. The corresponding data are presented in Figure 5d. The manuscript text has been adjusted to incorporate these new findings. These supplementary data have influenced our conclusions, as emphasized in the revised manuscript.

- The authors state in the abstract that the LDLR KO mice are “largely protected from CCHFV infection.” Yes, a reduction in viremia and virus loads in the liver and spleen were seen; however, if LDLR is such a crucial receptor, you would expect no virus replication by day 4. Furthermore, these mice were challenged with 400 PFU intraperitoneally of a highly mouse- adapted CCHFV strain, which is probably 100 times the LD50. A dose titration needs to beconducted to show that the effects seen here are not because of the significant overload of the animal with the virus, and a titration would be recommended titration of virus doses.

Response: We concur with the reviewer's perspective that LDLr is likely not the sole receptor. This aspect has been further elucidated in the revised manuscript. Consistent with the in vitro data, we observed a reduced level of infection in knockout (KO) mice. Our newly obtained data on the variability in the protein composition of CCHFV particles partially elucidates why the virus can still enter cells even in LDLR KO mice (figure 7 and 8). The decision to use 400 PFU was made based on our standard procedural protocol.

- Furthermore, an animal number of 6 per group with no repetitions is unacceptable, especially regarding reproducibility. these experiments need to be repeated with larger sample sizes. The macroscopic signs of disease listed here, for example, weakness, swollen eyes marked incoordination, and light bleeding around the marking hole in the ear, have not been described in any other CHF mouse model and need to be defined. How was weakness measured how were swollen eyes measured how was marked in coordination measured? is light bleeding around the marked hole of the year an artifact? Do the panels in Figure 5E come from the same WT/LDLR KO mouse? More pictures are unnecessary as the damage in the hepatic cells in the kale mice seems to be apparent, but it's difficult as a reader to judge how strong the impact really was on the liver and the spleen.

Response: We conducted additional experiments, effectively increasing the number of mice per condition. Regarding the signs of disease, the text has been revised accordingly (line 314-319 and extended figure 6). The images in Figure 5e are sourced from distinct mice. The histopathologist's report can be included as source data linked to this paper.

- Liver samples should have had immunohistochemistry. Liver damage isn't everything, For instance, CCHFV strain Hoti isn't lethal in mice, but causes damage in the liver.

Response: The presence of CCHFV was analysed by PCR and we highlighted the liver damage on the picture in Figure 5e.

Methods and Figures

290: define the ATCC strain number for SW-13

Response: Now added line 442

317: This pseudotype needs to be adequately characterized to show that conformationally dependent Gc is, in fact, incorporated on the surface of the virion. Additionally, characterization is needed of this model.

Response: Already answered above as a neutralization assay has been done (Extended data figure 1)

346: were thawn -> were thawed and infected

Response: Now modified line 505

347: Astonishingly, the authors were able to generate 5×10^8 ffu of an infectious dose from the pseudotype for a system that usually doesn't work well together (VSV and Gc/Gn assemble in different cell compartments). None of the VSV-CCHFV papers, even the one referenced in the paper, do not produce titers that high. It is just an inefficient pseudotype virus. This is highly suspect without additional characterization of this pseudotyped model.

Response: This comments has already been addressed above for the comments on the pseudotyped virus and the mistake in the reference is now corrected line 480

352: Justification in your methods approach as to why you used 500 million virus particles of your pseudotype, and only 500 virus particles of the actual wildtype virus seem very suspect without justification in this approach.

Response:

To ensure the infection of all susceptible cells before their rapid growth (haploid cells exhibit high growth rates), a high Multiplicity of Infection (MOI 10) is employed to enhance the likelihood of infecting all susceptible cells. Conversely, for validation purposes with the goal of investigating the impact of knockout (KO) on infection, and to avoid forcing the virus into cells, a lower MOI (here MOI 0.1) was utilized. This has been mentioned in revised MS line 506

361: If the actual stocks of the CCHFV virus were grown and amplified in SW-13 cells, why then were A549 and Vero's cells chosen as the haploid knockouts? Please justify the switch from a human adrenal carcinoma cell line to a monkey kidney cell line. Why were not liver cells are chosen, such as Huh-7 or HepG2 cells? Both of these liver cell lines play a role in hemorrhagic fever virus tropisms, LDL/LDLR, and support the growth of CCHFV to high titers. Please justify why liver cells were not used in these experiments.

Response:

Regarding diploid cells, SW13 was not employed as a knockout (KO) due to the challenges associated with knocking them out, and the same difficulty applied to Huh7. The decision not to use hepatoma cells does not diminish the significance of the findings- as our data were confirmed in human blood vessels. In the revised manuscript, we have also included HepG2 cells for a different purpose (figure 7).

Figure 3: The normalization of BRET scoring seems arbitrary. 1.00-1.02 changes do not seem profound at all, and this reviewer questions if any of this is noise without explaining how this readout is supposed to be interpreted in results. Because in subsequent panels, the changes are on the orders of 1.2-1.4, yet both are supposed to be significant changes? Please justify what these results mean in this context.

Response: BRET is the ratio of acceptor to donor counts. Over the years, different donor and acceptor pairs have been engineered to decrease noise and increase signal. As a BRET donor, Nluc usually produces smaller increases in BRET with a cleaner signal- to-noise, whereas Rluc typically has larger responses that can however be more sensitive to substrate, BRET acceptor and dipole-dipole orientations. As such, it is not meaningful to compared the raw BRET ratios or the normalized responses between assays with different donor-acceptor pairs (and different biological questions), specifically the binding assay which requires an N-terminally tagged LDLR (using Nluc) and the internalization assay that used Rluc. Notably, we have presented all data with the appropriate positive and negative controls and applied statistical analysis throughout (Figure 3).

Figure 5D: the axis in the serum is off compared to the others in this panel. Why include 10^{-5} as the x-axis in that panel, but 10^3 in the others, please address this or correct it.

Response: Now modified Figure 5c

Referee #2 (Remarks to the Author):

Monteil et al. conducted a study to investigate the role of low-density lipoprotein receptor (LDLR) as a critical entry receptor for Crimean-Congo Hemorrhagic Fever Virus (CCHFV) by directly binding to the CCHFV glycoprotein Gc. The authors utilized haploid cell screening techniques to identify LDLR as a potential receptor candidate and found that disruption of LDLR expression prevented the virus from infecting cells. Specifically, the study elucidated the mechanism by which Gc causes endocytosis of the LDLR and highlighted the inhibitory effect of soluble LDLR as a decoy against viral infection.

Additionally, the infectivity was assessed in organoid-derived pericyte and endothelial cells, as well as LDLR knockout mice, offering potential targets for antiviral strategies. However, certain questions still need to be addressed to enhance the overall understanding of LDLR's role as a cellular entry receptor for CCHFV. In addition, there are internally contradictory results within the manuscript.

Major comments:

1. Both Gc and Gn are glycoproteins that may play roles in viral entry. The researchers utilized a glycoprotein pseudotyped virus for their screening experiments. Based on the information provided in the Methods section, both Gn and Gc were present on VSV- CCHFV_G. However, the manuscript lacks a clear distinction between these two glycoproteins, with only Gc being emphasized. Therefore, it is crucial to include compelling evidence that Gn is NOT involved in LDLR receptor-mediated viral entry or shows no- association with LDLR receptor.

Alternatively, the inclusion of Gn as a negative control in the main experiments should also be necessary.

Response: We agree- We added Gn as well as Gn-Gc in the BRET analysis and we also preformed affinity experiments (Figure 3)

2. In Fig. 1C, there were 4 clones that showed near complete abolition of viral infection. Although authors mentioned about other clones may reside at regulatory gene region, it is better to include KO sequence information for clones in the supplement data.

Response:

For the remaining clones, we lack information on knockout (k.o) sequences. We speculate that alterations may exist in regulatory sequences not covered by the whole exome capture, thereby escaping our detection. This explanation is now provided in the text (line 138).

3. According to Fig. 1d, each of the cell line contained different *ldlr* mutant separately. Will these mutations lead to changes in LDLR expression levels or induce alternative splicing events? The authors should investigate the specific alterations occurring in LDLR at both the transcriptional and protein levels, which could be responsible for the decrease in viral infection. **Response: Yes, indeed the 3 clones carry different alterations in the LDLR gene, which are predicted to induce a loss-of-**

19fuction. Based on this finding we went on to validation studies using engineered cells with full LDLR k.o. Our data demonstrate that a loss of LDLR protein protects from infection. We could analyze the clones from the screen in more detail, but we typically only use the screen to generate a hypothesis which is validated in 'cleaner' cell lines and model systems than the randomly mutagenised cell clones (as we did in the present manuscript).4. As depicted in Fig. 2, *Ildl* KO in both cell lines did not confer complete resistance to virus infection. Authors should address this issue.

Response: We have now acknowledged this fact in revised MS and, furthermore, demonstrated that cell-derived proteins on the virus surface can influence the entry of the virus into new cells (figure 7 and 8).

5. In Fig. 2, in addition to knocking out *Ildl*, the authors could also generate point mutants of *Ildl* based on the results shown Fig.1d. These mutants can be used to conduct gain-of-function studies to further explore their impact on CCHFV infection.

Response: Yes, we could recapitulate the exact point mutations, but since 2 out of 3 lead to a premature stop, it is likely that any mutation causing a loss-of-function of the LDLR gives rise to CCHFV resistance (as demonstrated in the k.o. validation experiments). Therefore we cannot base a kind of structure-activity relation study on the results from the screen, we do not get information on the exact part of LDLR binding to CCHFV glycoproteins.

6. (Fig. 2) In addition to utilizing qPCR as the sole method, the authors could consider incorporating other data to strengthen their findings. For example, they could investigate the reduction of GFP signal in a LDLR knockout cell line upon infection with the pseudotyped virus.

Response: IFA data are now added as Extended data figure 4.

7. (Fig. 3) The current BRET assay configuration does not provide conclusive evidence to establish a direct binding between CCHF Gc and LDLR. The fluorescent and bioluminescent tags used in the assay can interact within a range of 10 nm, which is relatively long at the cellular level. Therefore, in addition to the BRET assay, it is crucial to gather further data to fully support the hypothesis: (a) assays that could show direct interactions: such as IP, PLA, purified protein SPR, etc. (b) functional assays: such as viral attachment/internalization at different temperatures -> qPCR.

Response: We now added affinity data between recombinant Gc, Gn, Gn-Gc with LDLR. To assess the attachment part of the virus to LDLR, as it is difficult to obtain a high infectious titer of CCHFV for a binding assay at 4°C, we runned a competition assay between CCHFV and LDL, the natural ligand of LDLR at the surface of the cells. Data are now presented in figure 3e-f

8. Based on Fig. 3, the authors employed BRET to demonstrate the interaction between Gc and LDLR, without mentioning Gn. As Gn does not bind to LDLR, it could be a good negative control. Furthermore, the authors should provide clarification regarding the specific regions of LDLR that are involved in the interaction with Gc. Additionally, it would be valuable for them to elucidate the

21binding affinity responsible for the Gc-LDLR interaction. These details would help provide comprehensive understanding of the molecular mechanisms underlying this interaction.Response: Gn is now added to the data, as well as Gn-Gc and also affinity measurement (figure 3)

9. According to Fig. 3, the authors demonstrated that the N-terminal of LDLR binds with Gc, while the C-terminal domain traffics to the early endosome upon Gc stimulation. However, they did not specify whether the N-terminal binding is essential for C-terminal endocytosis. Further clarification is needed regarding the requirement of N-terminal binding for C-terminal endocytosis.

Response: LDLR is a protein that has a constant natural recycling cycle in the cell, being endocytosed by some cellular secreted proteins (like PCSK9) and degraded or recycled to the cell surface. Meaning the N-Terminal binding of Gc is not essential for C-Terminal endocytosis as other proteins can lead to this.

10. As shown in Fig. 4, soluble LDLR, primarily in its N-terminal form, acts as a decoy for the receptor itself. Similarly, does LDL compete with Gc binding to LDLR?

Response: A competition assay in BRET as well as CCHFV/LDL competition assay in cell culture are now added to the MS (Figure 3)

11. (Fig. 4) It has been reported in some studies that LDLR undergoes cleavage by PCSK9, as well as Bone Morphogenetic Protein 1 (BMP1), among others. The mechanisms underlying the formation of soluble LDLR, and its inhibition mechanism of viral infection remain unclear. The authors could discuss this topic.

Response: Thanks for this comments. Yes it is an interesting field but as you mentioned little is known about it. It will be interesting in a future study to check the role of endogenous sLDLR in the course of infection in human.

12. In Fig. 5, the authors performed infections of organoid-derived pericyte and endothelial cells with CCHFV. However, it is not clearly stated how they quantified the percentage of infected cells and compared the infectious rates across different cell types. It is important for the authors to provide detailed information regarding the methodology employed to determine the percentage of infected cells and to compare the infectious rates among the various cell types. **Response: The level of infection was measured in the pool of cells by qRT-PCR. We didn't check which type of cells are infected with CCHFV neither at which level. It will be studied in the future.**

Minor comments:

1. In Fig 2, the authors should provide an explanation for their choice of clones C2, C12, and C8, C10, C11.

Response: The clones were not chosen *per se*. The validation of the KO by flow cytometry gave 2 clones KO for Vero cells and 3 in A549. We kept their original number in the figure as we could have renamed them C1, C2 and C1, C2, C3.

2. Why would the authors choose to use IFNR blocking, other than A129 mice?

Response: because A129 mice KO for LDLR are not commercially available

3. Most figures lack titles and legends, making it difficult for readers to comprehend the data.

Additionally, the authors should use more distinct colors to differentiate between each population, especially for BRET assay. The authors could enhance the readability and of the figures in their work.

Response: It is now modified to be more readable

4. Some figures depict significant differences using asterisks (Figures 3b, d), while others use raw p-values. The authors should maintain uniformity throughout the paper, and asterisks may provide a more straightforward representation.

Response: We agree about the asterisks making the significance easier to read. Now modified

5. In the Introduction section, line 66, it would be preferable to include the specific name of the tick species that serves as a CCHFV reservoir.

Response: Agree, Now added line 96

6. Line 183: This sentence should be moved or modified (e.g., "These data indicate that soluble LDLR can prevent CCHFV infections").

Response: now modified line 273

7. Line 202-203: it is not a title.

Response: Thanks for highlighting this mistake. It is now modified line 295

8. Line 410: The tubes were then incubated at 37°C with shaking for 30 minutes.

Response: It is now corrected line 570

Referee #3 (Remarks to the Author):

Monteil et al. report the identification of the low density lipoprotein related receptor (LDLR) as a cellular receptor for Crimean Congo hemorrhagic fever virus (CCHFV). The authors used a genetic screen to identify LDLR as a candidate receptor. They then used a series of assays, including work with recombinant proteins, VSV pseudotypes, infectious viruses (lab and clinical isolates), and LDLR knockout mice to support their claim for the role of LDLR as a cellular receptor for CCHFV. Given the importance of CCHFV as an emerging human pathogen and the virus' epidemic potential, the work is significant, and the findings are of broad interest. The work is also exciting because of the increasingly recognized role of LDLR-related proteins (alluded to in the manuscript) as receptors for many viruses for which cellular receptors had previously remained elusive.

25However, certain issues would need to be addressed for the work to meet the standard usually required to make a strong claim that a given protein is a bona fide cellular receptor that also contributes to pathogenesis. This includes showing that receptor expression allows for the attachment and internalization of viruses or virus-like particles, which is a critical piece of information that is currently missing from the manuscript. There are also concerns about the design and execution of the in vivo study, as noted below.

Major

1. In regards to the VSV pseudotype, the authors write in lines 96-98, “This virus lacks the region coding for any glycoproteins, and therefore produces non-infectious particles unless reconstituted with a novel surface glycoprotein, i.e. in our screen with the Gc glycoprotein of the CCHFV.” This phrasing suggests that only Gc is on the pseudotype. It seems unlikely that VSV pseudotype that contains Gc only on the surface would produce infectious particles, as the Gn/Gc heterodimer would be required. Could the authors clarify? (e.g., Figure 1a shows

that M was transfected when making the VSV pseudotype, but the methods mention a vector that encodes Gn and Gc was used). It would also be useful for the authors to describe what is known about envelope glycoprotein organization for CCHFV (how Gn assembles with Gc) somewhere in the manuscript.

Response: Our VSV-CCHFV contain both Gc and Gn. This is now corrected and we added neutralization data (extended data figure 1)

2. In a related point, the authors do not show that LDLR mediates virus binding to cells and internalization; this was only shown with recombinant Gc in isolation. Gc (in isolation) is not what is found on the virus and would assemble as a heterodimer with Gn on the virion surface (to stay in the pre-fusion conformation). It's unclear whether Gc in isolation would be in the right conformation to bind receptors. Assays should be done with virus-like particles/pseudotype that contains both Gn/Gc. The authors, for example, could demonstrate that their pseudotype contains Gn and Gc through SDS-page gels of purified preparations and show that sLDLR binds pseudotypes in ELISAs.

Response: We now added data for BRET Gc+Gn as well as an affinity study. We also runned a competition assay between CCHFV and LDL, the natural ligand of LDLR at the surface of the cells. (Figure 3)

3. Additional information should be provided about the Gc preparation that was obtained commercially. Is this Gc post-fusion trimer, or a monomeric form of Gc?

Response: It is a monomeric form

4. "6xHis-tagged CCHFV Gn" is mentioned in the methods as having been used in the BRET assays. Was Gn tested in these assays for LDLR binding? Showing that Gc but not Gn binds a cell surface receptor for CCHFV would have significant implications to our understanding of the entry of this virus (and of related viruses that share similar spike protein organization, e.g., containing Gn and Gc).

Response: This data are now added to the MS (Figure 3)

5. As for the animal studies, it is unclear why the authors sacrificed all mice at day 4 days post-infection rather than waiting for the mice to meet euthanasia criteria and only sacrificing a pre-determined subgroup of the wild-type mice for virological analysis. This is important because while mean viral loads are suppressed in the knockout animals, there is still viral replication in these animals, with some of the mice having tissue viremia comparable in magnitude to that of wild-type animals. Is it possible that the knockout animals would nonetheless succumb to infection at a later time point? A study by the same group (PMID 20164263) observed interferon knockout mice for 11

27days. The more extended time period would clarify whether LDLR-independent replication results in pathogenesis in the knockout mice.

Response: Agree. We sacrificed KO mice at the same time than the wild-type to be able to compare viremia. In the second mice study in the revised the MS, we kept a group of KO mice for survival study. We show the data as figure 5d. And the text was modified accordingly the new data (line 297 to 332)

6. Some of the disease phenotypes are noted in the text (“macroscopic signs of diseaseweakness, swollen eyes, marked incoordination, piloerection, light bleeding around the marking hole in the ear”), but this is only mentioned in passing with no quantification, or scoring scheme, or objective metric to follow how frequent these features were observed in knockout vs. wild-type mice.

Response: The disease phenotype was observed only in wild type mice. The decision of euthanasia of sick mice is done following a scoring system fixed by Karolinska Institutet. This point is now addressed in the MS (line 307) and the scoring is presented in Extended data figure 6

7. Also, it seems that rather than weighing the animals daily, the authors only weighed the animals at a single time point. Could the authors clarify why the animals were only weighed once?

Response: Weighting animals daily in our BSL4 lab means to make them go under anaesthesia every day as the mice can be handled only when they are sleeping. So many anaesthesia is deleterious for the mice health and can biased the data of infection.

8. Figure 5E. Histopathology should have included stains for viral antigen or RNA; this would clarify whether the changes noted in the tissues are a result of viral replication, and would also allow for assessment of the general amount of viral antigen/RNA in tissues for the wild-type vs knockout animals. Are these images each from one wild-type and one knockout mouse? Histopathology would have to be performed on multiple mice with images provided in the extended data for the observation to be rigorous.

Response: Histopathology was performed on several mice. The pictures can be furnished as well as the report from histopathologist. To note, the general amount of viral RNA in tissues for the wild type vs KO mice is showed in figure 5, as RT-PCR is more sensitive than IF.

Overall, a more thorough assessment would be required to make the claim that the knockout animals are “largely protected from the disease,” in addition to viral RNA levels that are reduced but not absent in the knockout animals.

Response: Thank you for this comments. New animal experiment is now included and the text is modified accordingly (figure 5 c-d and line 314-319)

Minor

1. What strain of RVFV was used in assays? This should be noted in the methods.

Response: Agree. The strain is now noted line 457

2. Figure 4 – labeling virus names in the panels would be helpful to readers.

29Response: Agree. Now done Figure 4 and 7

3. Additional relevant literature that could be cited (lines 260-264) is the identification of LDLR-related proteins as receptors for Oropouche virus (PMID: 35939689) and certain alphaviruses (PMID: 35939689, 34929721)

Response. Agree but we are limited in number of citations.

Decision Letter, first revision:

Message:

Our ref: NMICROBIOL-23071732A

21st February 2024

Dear Ali,

Thank you for your patience as we've prepared the guidelines for final submission of your Nature Microbiology manuscript, "LDLR as an important receptor for Crimean-Congo Hemorrhagic Fever virus" (NMICROBIOL-23071732A).

I am very sorry for the delay, since we encountered issues with our processes.

Please carefully follow the step-by-step instructions provided in the attached file, and add a response in each row of the table to indicate the changes that you have made. Please also check and comment on any additional marked-up edits we have proposed within the text. Ensuring that each point is addressed will help to ensure that your revised manuscript can be swiftly handed over to our production team.

In recognition of the time and expertise our reviewers provide to Nature Microbiology's editorial process, we would like to formally acknowledge their contribution to the external peer review of your manuscript entitled "LDLR as an important receptor for Crimean-

30Congo Hemorrhagic Fever virus". For those reviewers who give their assent, we will be publishing their names alongside the published article.

Nature Microbiology offers a Transparent Peer Review option for new original research manuscripts submitted after December 1st, 2019. As part of this initiative, we encourage our authors to support increased transparency into the peer review process by agreeing to have the reviewer comments, author rebuttal letters, and editorial decision letters published as a Supplementary item. When you submit your final files please clearly state in your cover letter whether or not you would like to participate in this initiative. Please note that failure to state your preference will result in delays in accepting your manuscript for publication.

Cover suggestions

COVER ARTWORK: We welcome submissions of artwork for consideration for our cover. For more information, please see our guide for cover artwork.

Nature Microbiology has now transitioned to a unified Rights Collection system which will allow our Author Services team to quickly and easily collect the rights and permissions required to publish your work. Approximately 10 days after your paper is formally accepted, you will receive an email in providing you with a link to complete the grant of rights. If your paper is eligible for Open Access, our Author Services team will also be in touch regarding any additional information that may be required to arrange payment for your article.

Please note that *Nature Microbiology* is a Transformative Journal (TJ). Authors may publish their research with us through the traditional subscription access route or make their paper immediately open access through payment of an article-processing charge (APC). Authors will not be required to make a final decision about access to their article until it has been accepted. Find out more about Transformative Journals

Best regards,

Reviewer #1:

Remarks to the Author:

Without adding anything further, the authors have satisfactorily addressed all of my concerns.

Reviewer #2:

Remarks to the Author:

The authors have adequately addressed prior concerns. Importantly, the animal studies demonstrate that while LDLR may play an important role as an entry receptor that is required during CCHFV pathogenesis, other receptors/alternative entry pathways are also probably at play, given that delayed death was observed in LDLR-deficient mice.

I have the following comments related to data analysis and statistics:

MAJOR:

1. Extended Data Figure 1 – Analysis should be performed to confirm statistical significance of the neutralization curves.
2. Extended Data Figure 3b. Related to the text, "...the binding affinity of GP38, a secreted CCHFV glycoprotein (GP38) of unknown function that is the target of protective antibodies, was 1000-fold lower (1.8 μ M) than the affinities of LDL or Gc for LDLR."

The binding data shown has very low signal. Are the authors sure that the detected binding is real? Because of how the data are presented, it is impossible to discern whether the signal was detected at different concentrations, or only at the highest concentration. If the calculated affinity is 1.8 μ M, a concentration higher than 26 nM should have been tested to determine if the response is increased in this assay. Without additional experiments, the biological significance of this very low response is unclear (are the authors sure that a weak signal does not represent "no binding"?).

3. Extended Data 4b – the IFA data shown should be quantified with statistics provided.
4. Figure 5d – Survival course of wild type and LDLR KO mice – Statistical analysis should be run between the groups to determine whether the observed difference in survival times between the groups is statistically significant.

MINOR:

In the abstract,

"the cellular human receptor and the host cell factors essential for Crimean-Congo hemorrhagic fever virus (CCHFV) infection have not yet been identified"

Suggest rephrasing as "the cellular human receptor" implies that CCHFV would have only

one receptor.

Reviewer #3:

Remarks to the Author:

Major comments

The authors have addressed most of our inquiries, and we are generally content with the revised data. However, unfortunately, a very similar paper has been recently published on January 5th this year in Cell Research (<https://doi.org/10.1038/s41422-023-00917-w>). This paper also asserts LDLR as the crucial entry receptor for CCHFV and provides more comprehensive supporting data in terms of (1) viral infection assays, (2) LDLR domain mapping, and (3) the use of anti-LDLR as a treatment in a mouse model.

In comparison, despite considerable overlaps, this manuscript may offer more detailed binding and endocytosis quantification data, an organoid model, supportive data from clinical samples, and a thorough comparison between lab-generated and clinically isolated CCHFV. Additionally, it explores the potential secondary receptor during the human infection process.

Both manuscripts provide unique data that mutually reinforce their respective findings. To further enhance the novelty and emphasize distinct aspects in this manuscript, the authors might consider incorporating additional experiments and points of view.

Minor comments:

1. While we appreciate the added IFA data in Figure 2 as feedback, we would still like to see more parallel data, particularly beyond qPCR. The authors could consider conducting experiments such as western blot of viral proteins, plaque assays, etc. It's worth noting that the CR paper includes these types of data, supporting this manuscript. Since the information from the CR paper addresses our concerns, there may be no further needs to request similar experiments here again.
2. Similarly, we would like to see some parallel data for the endocytosis part other than BRET assay. For example, the author could label the virion particles with two lipophilic probes, which could detect the endocytosis process during infection.
3. Line 4: there are two commas at the end.
4. Line 209: please rephrase "taken together" since it's right after line 207 "taken together".
5. Line 214: the title is not in the new line.
6. Line 236: need a blank space after the period.
7. Line 277: need a blank space after the period.

Author Rebuttal, first revision:

Answer to referees

Reviewer #1 (Remarks to the Author):

33Without adding anything further, the authors have satisfactorily addressed all of my concerns.

Thank you for all your comments that helped to strongly improve the manuscript

Reviewer #2 (Remarks to the Author):

The authors have adequately addressed prior concerns. Importantly, the animal studies demonstrate that while LDLR may play an important role as an entry receptor that is required during CCHFV pathogenesis, other receptors/alternative entry pathways are also probably at play, given that delayed death was observed in LDLR-deficient mice.

I have the following comments related to data analysis and statistics: MAJOR:

1. Extended Data Figure 1 – Analysis should be performed to confirm statistical significance of the neutralization curves.

Thank you for this comments. It is now addressed and the legends has been modified accordingly.

2. Extended Data Figure 3b. Related to the text, "...the binding affinity of GP38, a secreted CCHFV glycoprotein (GP38) of unknown function that is the target of protective antibodies, was 1000-fold lower (1.8 μ M) than the affinities of LDL or Gc for LDLR."

The binding data shown has very low signal. Are the authors sure that the detected binding is real? Because of how the data are presented, it is impossible to discern whether the signal was detected at different concentrations, or only at the highest concentration. If the calculated affinity is 1.8 μ M, a concentration higher than 26 nM should have been tested to determine if the response is increased in this assay. Without additional experiments, the biological significance of this very low response is unclear

(are the authors sure that a weak signal does not represent "no binding"?).

We thank the reviewer for this comment. We did in fact test concentrations higher than 26 nM (33, 66, 132 and 263 nM) which are correctly annotated to the right of the QCM data. The affinity is 0.18 μ M (Typo mistake), which is 100-fold lower than the affinities of LDL or Gc for LDLR. We have corrected the text accordingly (line 193).

3. Extended Data 4b – the IFA data shown should be quantified with statistics provided.

Thank you for this comments. It is now addressed and the legends was modified accordingly.

4. Figure 5d – Survival course of wild type and LDLR KO mice – Statistical analysis should be run between the groups to determine whether the observed difference in survival times between the groups is statistically significant.

Thank you for this comments. It is now addressed and the legends was modified accordingly.

MINOR:

In the abstract,

“the cellular human receptor and the host cell factors essential for Crimean-Congo hemorrhagic fever virus (CCHFV) infection have not yet been identified”

Suggest rephrasing as “the cellular human receptor” implies that CCHFV would have only one receptor.

Thank you for this comment. It is now modified line 63

Reviewer #3 (Remarks to the Author):

Major comments

The authors have addressed most of our inquiries, and we are generally content with the revised data. However, unfortunately, a very similar paper has been recently published on January 5th this year in Cell Research (<https://doi.org/10.1038/s41422-023-00917-w>). This paper also asserts LDLR as the crucial entry receptor for CCHFV and provides more comprehensive supporting data in terms of (1) viral infection assays, (2) LDLR domain mapping, and (3) the use of anti-LDLR as a treatment in a mouse model.

In comparison, despite considerable overlaps, this manuscript may offer more detailed binding and endocytosis quantification data, an organoid model, supportive data from clinical samples, and a thorough comparison between lab-generated and clinically isolated CCHFV. Additionally, it explores the potential secondary receptor during the human infection process.

Both manuscripts provide unique data that mutually reinforce their respective findings. To further enhance the novelty and emphasize distinct aspects in this manuscript, the authors might consider incorporating additional experiments and points of view.

Thank you for your comment. We agree that the paper published in CR support our findings and that we go further in exploring clinical samples, virus produced on tick cells as well as the potential other receptors. We will investigate the role of LDLR and co receptor in future studies..

35Minor comments:

1. While we appreciate the added IFA data in Figure 2 as feedback, we would still like to see more parallel data, particularly beyond qPCR. The authors could consider conducting experiments such as western blot of viral proteins, plaque assays, etc. It's worth noting that the CR paper includes these types of data, supporting this manuscript. Since the information from the CR paper addresses our concerns, there may be no further needs to request similar experiments here again.

Thank you for this comment. In our hands, IFA is more precise than Western-blot to show the lower amount of nucleoprotein (NP) as we see individual cells data instead of a pool. We believe that our data (PCR and IFA) should be sufficient to show that the KO of LDLR impact CCHFV infection.

2. Similarly, we would like to see some parallel data for the endocytosis part other than BRET assay. For example, the author could label the virion particles with two lipophilic probes, which could detect the endocytosis process during infection.

Thank you for this comment. The aim of the BRET was to show the attachment of the viral glycoproteins to LDLR. In revised MS, we added competition data with LDL to the MS to show that the virus attach to LDLR at the surface of the cells. The labeling CCHFV is not an easy tasks, due to that the purification of virus is very complicated as well as the concentration of virus is not too much, and we do not believe this experiment will bring much more information.

3. Line 4: there are two commas at the end.

Now modified line 4

4. Line 209: please rephrase "taken together" since it's right after line 207 "taken together".

Now Modified line 211

5. Line 214: the title is not in the new line.

Now modified line 217

6. Line 236: need a blank space after the period.

Now modified line 239

7. Line 277: need a blank space after the period.

Now modified line 280

Final Decision Letter:

Message: 11th March 2024

Dear Ali and Vanessa,

I am delighted to accept your Article "Crimean-Congo Hemorrhagic Fever virus uses LDLR to bind and enter host cells" for publication in Nature Microbiology. Thank you for having chosen to submit your work to us and many congratulations.

We will publish your paper on an accelerated schedule. Please carefully review the details below and contact us immediately at microbiology@nature.com if you have any travel plans or other conflicts that may make you unable to respond to us for the next 5-7 days.

In approximately 2 business days you will receive a link to choose the appropriate publishing options for your paper and complete the appropriate grant of rights necessary to publish your work. As it is vital that this process not be delayed, we strongly encourage you to Find out more about Transformative Journals

Authors may need to take specific actions to achieve compliance with funder and institutional open access mandates. If your research is supported by a funder that requires immediate open access (e.g. according to Plan S principles) then you should select the gold OA route, and we will direct you to the compliant route where possible. For authors selecting the subscription publication route, the journal's standard licensing terms will need to be accepted, including self-archiving policies. Those licensing terms will supersede any other terms that the author or any third party may assert apply to any version of the manuscript.

If you have any questions about our publishing options, costs, Open Access requirements, or our legal forms, please contact ASJournals@springernature.com.

2If you have not already done so, we strongly recommend that you upload the step-by-step protocols used in this manuscript to the Protocol Exchange. Protocol Exchange is an open online resource that allows researchers to share their detailed experimental know-how. All uploaded protocols are made freely available, assigned DOIs for ease of citation and fully searchable through nature.com. Protocols can be linked to any publications in which they are used and will be linked to from your article. You can also establish a dedicated page to collect all your lab Protocols. By uploading your Protocols to Protocol Exchange, you are enabling researchers to more readily reproduce or adapt the methodology you use, as well as increasing the visibility of your protocols and papers. Upload your Protocols at www.nature.com/protocolexchange/. Further information can be found at www.nature.com/protocolexchange/about.

Congrats again to you and your co-authors! I'm looking forward to seeing your paper published.

Best wishes,